# The bromodomain containing protein BRD-9 orchestrates RAD51–RAD54 complex formation and regulates homologous recombination-mediated repair

Qin Zhou[1,9], Jinzhou Huang[1,9], Chao Zhang[1,9], Fei Zhao[1], Wootae Kim[1], Xinyi Tu[1], Yong Zhang[1,2], Somaira Nowsheen[3,4], Qian Zhu[5], Min Deng[1], Yuping Chen[5,6], Bo Qin[1], Kuntian Luo[1,5,6], Baohua Liu [7], Zhenkun Lou [1], Robert W. Mutter[8✉] & Jian Yuan[5,6✉]

Homologous recombination (HR) is important for error-free DNA double strand break repair and maintenance of genomic stability. However, upregulated HR is also used by cancer cells to promote therapeutic resistance. Therefore, inducing HR deficiency (HRD) is a viable strategy to sensitize HR proficient cancers to DNA targeted therapies in order to overcome therapeutic resistance. A bromodomain containing protein, BRD9, was previously reported to regulate chromatin remodeling and transcription. Here, we discover that following DNA damage, the bromodomain of BRD9 binds acetylated K515 on RAD54 and facilitates RAD54's interaction with RAD51, which is essential for HR. BRD9 is overexpressed in ovarian cancer and depleting BRD9 sensitizes cancer cells to olaparib and cisplatin. In addition, inhibitor of BRD9, I-BRD9, acts synergistically with olaparib in HR-proficient cancer cells. Overall, our results elucidate a role for BRD9 in HR and identify BRD9 as a potential therapeutic target to promote synthetic lethality and overcome chemoresistance.

[1] Department of Oncology, Mayo Clinic, Rochester, MN 55905, USA. [2] Department of Radiation Oncology, Hubei Cancer Hospital, Tongji Medical College, Huazhong University of Science and Technology, Wuhan 430022, China. [3] Mayo Medical Scientist Training Program, Mayo Clinic School of Medicine and Mayo Clinic Graduate School of Biomedical Sciences, Mayo Clinic, Rochester, MN 55905, USA. [4] Department of Molecular Pharmacology and Experimental Therapeutics, Mayo Clinic, Rochester, MN 55905, USA. [5] Research Center for Translational Medicine, East Hospital, Tongji University School of medicine, Shanghai 200120, China. [6] Key Laboratory of Arrhythmias of the Ministry of Education of China, East Hospital, Tongji University School of Medicine, Shanghai 200120, China. [7] Guangdong Key Laboratory of Genome Stability and Human Disease Prevention, Department of Biochemistry & Molecular Biology, School of Basic Medical Sciences, Shenzhen University Health Science Center, Shenzhen 518055, China. [8] Department of Radiation Oncology, Mayo Clinic, Rochester, MN 55905, USA. [9] These authors contributed equally: Qin Zhou, Jinzhou Huang, Chao Zhang. ✉email: Mutter.Robert@mayo.edu; yuanjian229@hotmail.com

Detection and repair of damaged DNA are integral for cell survival and accurate transmission of genetic information to progeny. Defects in the DNA damage response (DDR) pathway contribute to genomic instability and oncogenesis, and also render tumor cells more sensitive to DNA-damaging cancer therapy[1–3]. Homologous recombination (HR) is a critical DNA double-strand break (DSB) repair pathway that uses the homologous sequence of a sister chromatid in the S and G2 phases of the cell cycle as a model to carry out error-free repair[4,5]. The essential event during HR is RAD51 filament formation on single-stranded DNA that enables search of a homologous sequence on the intact DNA strand[6–8]. In *Saccharomyces cerevisiae* and mammalian cells, Rad54, a SWI2/SNF2 chromatin-remodeling protein that can mediate the mobilization of nucleosomes and DNA-associated proteins[9], partners with Rad51 in its DNA strand exchange activity[10]. According to the current model, RAD54 extends and stabilizes Rad51–dsDNA filament, while simultaneously removing Rad51 from DNA once recombination has been initiated[11]. However, the detailed mechanism of how the RAD54–RAD51 complex carries out its function in the DDR has not been elucidated.

Bromodomains (BRDs) are evolutionarily conserved protein–protein interaction modules with diverse catalytic and scaffolding functions in a wide range of proteins and tissues. A well-known bromodomain function is in gene expression regulation through selective recognition and binding to acetylated Lys residues. BRD-containing proteins are frequently dysregulated in cancer, and many cancer-causing mutations have been mapped to the BRDs of these proteins themselves[12]. However, the role of BRDs in cancer is still not clear.

Somatic mutations, present in cancer genomes, are the consequence of multiple endogenous and exogenous mutational processes[13]. Different mutational processes generate unique combinations of mutational signatures, across various cancer types[14]. Previous studies have suggested potential roles for BRD-containing proteins in DNA repair, such as BRD4 and ZMYND8[15,16]. Here, we overlay a bioinformatics mutational signature analysis of the TCGA database with an established functional readout of DNA double-strand break repair to screen BRD-containing proteins for potential roles in HR[17]. We identify BRD9 as a HR regulator that facilitates RAD54 and RAD51 functions in HR by serving as a bridge between the two proteins. Because BRD9 is overexpressed in ovarian cancer, and targeting BRD9 sensitizes ovarian cancer to PARP inhibition and cisplatin, we show that BRD9 is a promising target to overcome therapeutic resistance in this disease.

## Results

**BRD9 is required for HR activity.** Through analyses of the TCGA database, we found that mutation of six BRD-containing proteins is associated with high HR-associated mutation signatures (signature 3) ($p < 0.01$) in ovarian cancer (Fig. 1a, Table 1). To further evaluate the potential roles of BRD-containing proteins in HR, we knocked down 42 BRD-containing proteins in HCT-116 cells, and assessed for the ability of cells to repair a site-specific double-strand break by HR using the widely employed DR-GFP reporter assay. As shown in Fig. 1b, three of the six BRD-containing proteins with a high HR mutation signature, BRD9, ASH1L, and ZMYND8, positively regulated HR as assessed by the DR-GFP assay. We selected BRD9, which has not previously been associated with DDR, for initial detailed investigation. First, we sought to confirm our screening data by performing HR and NHEJ reporter assays in OVCAR8 (Fig. 1c, d) and U2OS cells (Supplementary Fig. 1A, B) with two different shRNAs targeting BRD9. We observed a marked deficiency in HR

in BRD9-KD cells, whereas no significant changes were observed in NHEJ activity. Furthermore, we found that the BRD9-specific inhibitor, I-BRD9[18], significantly reduced HR, but not NHEJ activity (Fig. 1e, f, Supplementary Fig. 1C). Depletion of BRD9 significantly sensitized U2OS cells to PAPRi (Supplementary Fig. 1G). Collectively, these data suggested that BRD9 positively regulates DNA DSB repair by HR.

The phosphorylation of the histone H2AX at Ser139 (γ-H2AX) is a marker of DSB. Ionizing radiation (IR)-induced γH2AX foci can be monitored using immunofluorescence, and the resolution of these foci over time following DNA damage correlates with the ability of cells to carry out DSB repair. The absence of BRD9 did not affect IR-induced γH2AX foci formation at 2 h post IR. However, between 8 and 24 h after IR, we found significantly delayed clearance of γH2AX foci in BRD9 KD OVCAR8 cells, suggestive of decreased functional DSB repair activity (Fig. 1g, h). We did not find significant changes in the cell cycle profile when we deleted the BRD9 gene (Supplementary Fig. 2A), suggesting that the observed effect was not due to an indirect effect of a change in the cell cycle profile. Based on our findings of decreased HR activity in BRD9 KD cells, we predicted that BRD9 KD may inhibit the formation of IR-induced RAD51 foci, another commonly employed functional readout of HR[19]. Surprisingly, BRD9 KD in OVCAR8 cells did not result in deficient recruitment of RAD51 to DNA damage sites at 8 h. Rather, RAD51 foci persisted at sites of DNA damage in BRD9 KD OVCAR8 cells (Fig. 1i, j). RAD51 was also found to persist in the chromatin fraction following BRD9 KD in U2OS cells (Supplementary Fig. 2B). Similarly, RAD51 foci were sustained at damage sites in OVCAR8 cells treated with the BRD9 inhibitor (Supplementary Fig. 2C), whereas 53BP1 and BRCA1 foci clearance was not affected by BRD9 KD (Supplementary Fig. 2D). These results suggested that BRD9 may regulate the dissociation of RAD51 from chromatin following DNA damage, but not the initial recruitment of RAD51 to DNA damage sites. Together, these data supported a potential role of BRD9 in DDR and HR repair.

**BRD9 is required for interaction and co-localization of RAD51 and RAD54.** RAD54 has previously been reported to facilitate the dissociation of RAD51 from DSB sites and to promote HR[20]. In addition, as has previously been shown, we observed that RAD51 and RAD54 co-localized to sites of DNA damage in OVCAR8 cells (Fig. 2a, d)[21,22]. Given that BRD9 KD delayed the resolution of IR-induced RAD51 foci, we sought to assess the impact of loss of BRD9 on RAD54 recruitment to DNA damage sites. As displayed in Fig. 2a–c and Supplementary Fig. 2D, KD of BRD9 significantly reduced IR-induced RAD54 foci formation. In addition, KD of BRD9 resulted in significantly reduced co-localization of RAD54 with RAD51 (Fig. 2a, d). It has been reported that RAD54 interacts with RAD51 by its N-terminal region in a species-specific fashion[22,23]. Given the above findings, we hypothesized that BRD9 might modify this interaction following DNA damage. Indeed, as shown in Fig. 2e, f, the interaction between RAD51 and RAD54 increased following IR, which is consistent with prior reports[21,24]. However, KD of BRD9 abolished the IR-induced upregulated interaction between RAD51 and RAD54. Collectively, these results suggested that BRD9 may be important for recruiting RAD54 to DSBs following DNA damage, possibly by promoting the RAD51–RAD54 interaction. In order to explore this possibility further, we developed a BRD9-specific antibody and observed that BRD9 was recruited to sites of DNA damage following IR, colocalizing with RAD51 (Fig. 2g). However, depletion of RAD51 abolished the foci formation of

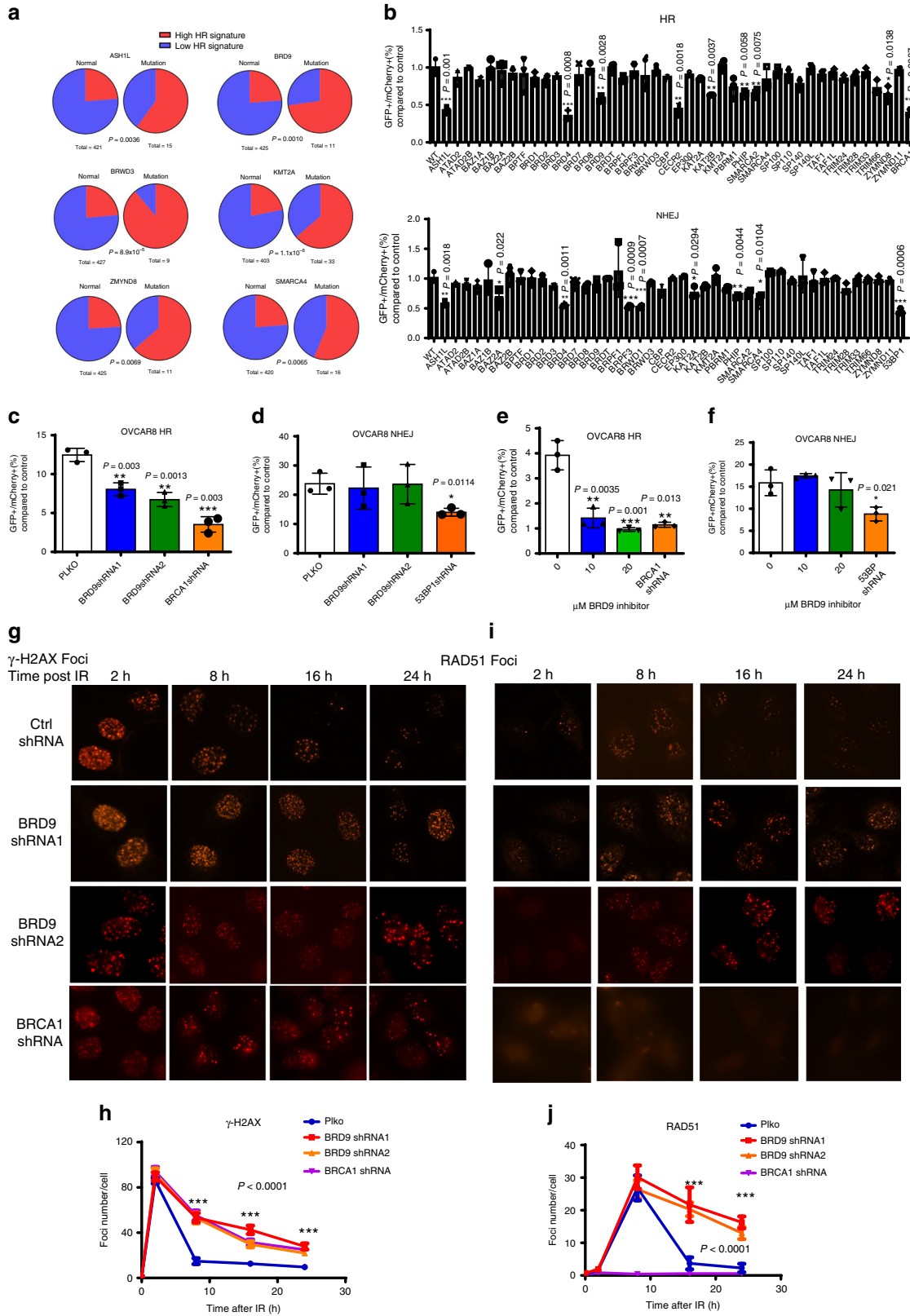

both BRD9 and RAD54 (Fig. 2g, Supplementary Fig. 2E). Taken together, these results suggested that RAD51 recruits BRD9, which in turn facilitates the recruitment of RAD54 to damage sites following IR.

**BRD9 separately interacts with RAD51 and RAD54 through its C-terminal and bromodomain.** We hypothesized that BRD9 might serve as a scaffolding protein for RAD51 and RAD54 during HR. To test this hypothesis, we performed

**Fig. 1 BRD9 is required for HR activity. a** Signature analysis of six BRD-containing proteins that have been associated with homologous recombination (HR)-related mutation signatures (signature 3) in ovarian cancer in the TCGA database. $p$ Values were calculated by one-sided Fisher Exact test. **b** Quantification of HR- and NHEJ-mediated DSB repair as assessed using GFP reporter assay in HCT-116 cells following knockdown of bromodomain-containing proteins. The indicated bromodomain-containing proteins were individually knocked down in HCT-116 cells transfected with GFP-tagged reporter plasmid. Thirty-six hours later, repair efficiency was assessed using flow cytometry. The BRCA1- and 53BP1-knockdown cells were used as positive control for HR and NHEJ, respectively. Representative data (mean ± SEM) are shown from $n = 3$ biologically independent samples; *$p < 0.05$, **$p < 0.005$, ***$p < 0.001$ by two-sided unpaired $t$ test. **c**, **d** Knockdown of BRD9 causes HR but not NHEJ deficiency. OVCAR8 cells were infected with lentivirus expressing the indicated BRD9 shRNAs. Thirty-six hours later, HR- (**c**) and NHEJ- (**d**) mediated repair capacity was assessed using flow cytometry. The BRCA1 and 53BP1 shRNAs were used as positive controls for HR and NHEJ, respectively. Representative data (mean ± SEM) are shown from $n = 3$ biologically independent samples. *$p < 0.05$, **$p < 0.01$, ***$p < 0.001$ by two-sided unpaired $t$ test. **e**, **f** BRD9 inhibitor (I-BRD9) selectively inhibits HR and not NHEJ activity. OVCAR8 cells were treated with 10 or 20 μM I-BRD9 for 36 h, and then subjected to HR (**e**) and NHEJ (**f**) assay as described in **c**, **d**. Representative data (mean ± SEM) are shown from $n = 3$ biologically independent samples. *$p < 0.05$, **$p < 0.01$, ***$p < 0.001$ by two-sided unpaired $t$ test. **g–j** Knockdown of BRD9 delays clearance of γ-H2AX and RAD51 foci. OVCAR8 cells were infected with lentivirus-expressing control (Ctrl) or BRD9 shRNA. Cells were exposed to 2-Gy irradiation and fixed at the indicated time points. Cells were stained for the indicated foci. **g**, **i** Representative images are shown of γ-H2AX and Rad51 foci after the indicated treatments and indicated time following 2-Gy irradiation. **h**, **j** γ-H2AX and Rad51 foci in OVCAR8 cells after the indicated treatment and time following 2-Gy irradiation were quantified. Representative data (mean ± SEM) are shown from three independent experiments. ***$p < 0.001$ by two-sided unpaired $t$ test. Scale bar, 10 μm.

### Table 1 Bromodomain-containing protein–HR signature.

| | Normal group | | | Mutation group | | | $p$ Value |
|---|---|---|---|---|---|---|---|
| | High signature | Low signature | High signature, % | High signature | Low signature | Percentage | |
| ASH1L | 100 | 321 | 23.75 | 9 | 6 | 60 | 0.003583 |
| ATAD2 | 104 | 325 | 24.24 | 5 | 2 | 71.43 | 0.01222 |
| ATAD2B | 107 | 326 | 24.71 | 1 | 2 | 66.67 | 0.1556 |
| BAZ1A | 105 | 325 | 24.42 | 4 | 2 | 66.67 | 0.0365 |
| BAZ1B | 104 | 323 | 24.36 | 5 | 4 | 55.55 | 0.04713 |
| BAZ2A | 108 | 326 | 24.88 | 1 | 1 | 50 | 0.4379 |
| BAZ2B | 107 | 309 | 25.72 | 2 | 8 | 20 | 0.7592 |
| BPTF | 106 | 320 | 24.88 | 2 | 8 | 20 | 0.4753 |
| BRD1 | 105 | 324 | 24.48 | 3 | 4 | 42.86 | 0.06896 |
| BRD2 | 109 | 323 | 25.23 | 0 | 4 | 0 | 1 |
| BRD3 | 109 | 325 | 25.11 | 0 | 2 | 0 | 1 |
| BRD4 | 106 | 322 | 24.77 | 3 | 5 | 37.50 | 0.3207 |
| BRD7 | 109 | 326 | 25.05 | 0 | 1 | 0 | 1 |
| BRD8 | 103 | 323 | 24.18 | 6 | 4 | 60 | 0.01851 |
| BRD9 | 101 | 324 | 23.76 | 8 | 3 | 72.72 | 0.00102 |
| BRDT | 107 | 325 | 24.76 | 2 | 2 | 50 | 0.2612 |
| BRPF1 | 107 | 326 | 24.71 | 3 | 3 | 50 | 0.2622 |
| BRPF3 | 106 | 324 | 24.65 | 4 | 3 | 57.14 | 0.06896 |
| BRWD1 | 103 | 320 | 24.35 | 6 | 7 | 46.15 | 0.0771 |
| BRWD3 | 101 | 326 | 23.65 | 8 | 1 | 88.89 | 8.90E−05 |
| CBP | 107 | 326 | 24.71 | 0 | 0 | 0 | 1 |
| CECR2 | 107 | 323 | 24.88 | 2 | 4 | 33.33 | 0.4667 |
| EP300 | 101 | 323 | 23.82 | 8 | 4 | 66.67 | 0.002421 |
| KAT2A | 108 | 326 | 24.88 | 1 | 1 | 50 | 0.4379 |
| KAT2B | N/A | N/A | N/A | 0 | 0 | 0 | 1 |
| KMT2A | 88 | 315 | 21.84 | 21 | 12 | 63.64 | 1.08E−06 |
| PBRM1 | 106 | 326 | 24.54 | 3 | 1 | 75 | 0.04997 |
| PHIP | 104 | 323 | 24.36 | 5 | 4 | 55.56 | 0.04713 |
| SMARCA2 | 104 | 323 | 24.36 | 5 | 4 | 55.56 | 0.04713 |
| SMARCA4 | 100 | 320 | 23.81 | 9 | 7 | 56.25 | 0.006474 |
| SP100 | 105 | 320 | 24.71 | 1 | 2 | 33.33 | 0.2851 |
| SP110 | 108 | 325 | 24.94 | 1 | 2 | 33.33 | 0.5719 |
| SP140 | 105 | 322 | 24.59 | 4 | 5 | 44.44 | 0.1636 |
| SP140L | 108 | 326 | 24.88 | 1 | 1 | 50 | 0.4379 |
| TAF1 | 106 | 320 | 24.88 | 3 | 7 | 30 | 0.4753 |
| TAF1L | 99 | 317 | 23.79 | 10 | 10 | 50 | 0.01197 |
| TRIM24 | 108 | 324 | 25 | 1 | 3 | 25 | 0.6851 |
| TRIM28 | 108 | 322 | 25.11 | 1 | 5 | 16.67 | 0.8241 |
| TRIM33 | 108 | 325 | 24.94 | 1 | 2 | 33.33 | 0.5791 |
| TRIM66 | N/A | N/A | N/A | 0 | 0 | 0 | 1 |
| ZYMND8 | 102 | 323 | 24 | 7 | 4 | 63.63 | 0.006857 |
| ZYMND11 | 109 | 326 | 25.06 | 0 | 1 | 0 | 1 |

The mutations of the genes with $p$ value < 0.05 caused the significant increase in HR mutation signature; threshold of high HR signature mutation: 75%; $p$ values were calculated by one-sided Fisher Exact test.

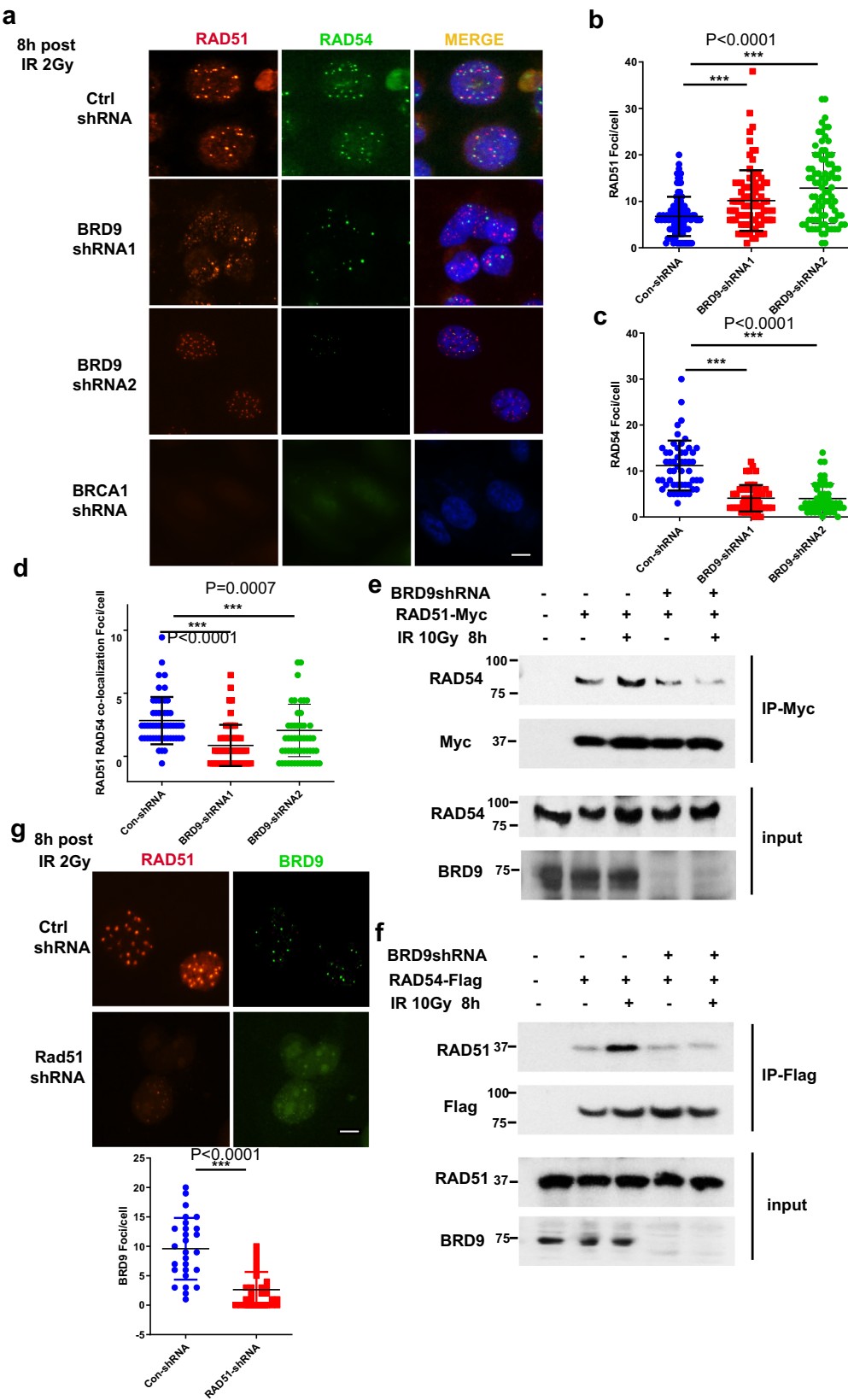

co-immunoprecipitation assays to examine the interaction between exogenous/endogenous BRD9 and RAD51/RAD54. As shown in Fig. 3a, b, BRD9 interacted with RAD51 constitutively in the presence or absence of DNA damage, whereas the interaction of BRD9 with RAD54 was induced following IR.

Next, we investigated which domain of BRD9 was important for its interaction with RAD51 and RAD54. To do so, we designed several BRD9 truncations (Fig. 3c) and performed co-immunoprecipitation experiments. As shown in Fig. 3d, the C terminus of BRD9 was important for the BRD9 interaction with

**Fig. 2 BRD9 is required for interaction and co-localization of RAD51 and RAD54. a–d** Representative immunofluorescence images of OVCAR8 cells demonstrating that BRD9 knockdown reduces RAD54 foci formation and RAD51/RAD54 co-localization. **a–c** OVCAR8 cells were infected with lentivirus-expressing control (Ctrl) or BRD9 shRNA and exposed to 2-Gy IR. Eight hours later, cells were stained with RAD51 (red) and RAD54 (green) antibodies. Representative immunofluorescence images are shown in **a**. RAD51 foci/cell (**b**), RAD54 foci/cell (**c**), and RAD51/RAD54 co-localized foci (**d**) after the indicated treatment were quantified. Representative data (mean ± SEM) are shown from three independent experiments. $n = 50$ cells examined over three independent experiments. ***$p < 0.001$ by two-sided unpaired $t$ test. Scale bar, 10 μm. **e, f** DNA damage-induced RAD51–RAD54 interaction is dependent on BRD9. Endogenous BRD9 was knocked out in HEK293T cells using shRNA, and the indicated constructs were expressed. Cells were exposed to 10-Gy irradiation. Lysates were collected after 8 h, and immunoprecipitation with the indicated antibodies was performed. Blots were probed with the indicated antibodies. **g** RAD51 is necessary for BRD9 foci formation. OVCAR8 cells were infected with lentivirus-expressing RAD51 shRNA, exposed to 2-Gy IR, and fixed after 8 h. Cells were stained with BRD9 (green) and RAD51 (red) antibodies. Representative images of the indicated foci are shown. BRD9 foci/cell were quantified. Representative data (mean ± SEM) are shown from three independent experiments. $n = 50$ cells examined over three independent experiments. ***$p < 0.001$ by two-sided unpaired $t$ test. Scale bar, 10 μm.

RAD51. In contrast, the BRD9 N terminus (bromodomain) was required for the interaction between BRD9 and RAD54 (Fig. 3d). In order to further understand the role of the BRD9 bromodomain in the BRD9 interaction with RAD54, we overexpressed BRD9 wild type (WT) or BRD9 containing a truncating deletion of the BRD9 bromodomain (Δ-Bromo) in 293T cells, and performed co-immunoprecipitation experiments (Fig. 3e). We found that both WT and Δ-Bromo BRD9 interacted with RAD51 constitutively with or without IR treatment. However, BRD9 WT, but not Δ-Bromo truncation, interacted with RAD54 following DNA damage, confirming that the BRD9 bromodomain is required for the interaction with RAD54.

Previous studies have reported that the small-molecule inhibitor, I-BRD9, inhibits BRD9 activity by directly binding to the BRD9 bromodomain[18]. We next sought to examine whether I-BRD9 affects the BRD9–RAD54 interaction. As shown in Fig. 3f, g, I-BRD9 abolished the DNA damage- induced BRD9–RAD54 interaction. Furthermore, we found that BRD9 WT, but not Δ-Bromo BRD9, promoted the RAD51 and RAD54 interaction following DNA damage (Fig. 3h). Taken together, these results suggested that BRD9 functions as a scaffolding protein for the interaction of RAD51 and RAD54 following DNA damage. Mechanistically, the C terminus of BRD9 constitutively binds to RAD51, while the N-terminal bromodomain of BRD9 binds to RAD54 in a DNA damage-inducible manner.

To further demonstrate the functional significance of BRD9 bromodomain, we reconstituted BRD9 KD cells with BRD9 WT or Δ-Bromo truncation. As shown in Fig. 3i, reconstitution of BRD9 WT but not Δ-Bromo BRD9 restored HR function, as assessed by the DR-GFP reporter assay. Furthermore, re-expression of BRD9 WT but not Δ-Bromo BRD9 reversed the hypersensitivity toward olaparib and cisplatin induced by BRD9 depletion (Fig. 3j, k). The persistence of RAD51 in the chromatin fraction following BRD9 KD also can be rescued by re-expression of BRD9 WT but not Δ-Bromo BRD9 (Supplementary Fig. 6A). Together, these results suggest that the BRD9 bromodomain is important for HR and chemosensitivity.

**RAD54 acetylation is important for BRD9 recognition and HR activity.** Bromodomains are protein domains that recognize acetylated lysine residues[25,26]. The finding that RAD54 interacts with the BRD9 bromodomain following IR led us to hypothesize that RAD54 may be acetylated following DNA damage. Indeed, as shown in Fig. 4a, RAD54 was acetylated in response to IR. In addition, we screened several acetyltransferases (Supplementary Fig. 3E) and deacetylases (Fig. 4c). We found that expression of acetyltransferases GCN5 and PCAF was positively related to RAD54 acetylation level (Fig. 4b, Supplementary Fig. 3E), while HDAC 6 and 11 were identified as the deacetylases of RAD54 (Fig. 4c). Interestingly, overexpression of GCN5 or PCAF, which

in turn enhanced RAD54 acetylation, dramatically increased BRD9–RAD54 interaction, while, reducing RAD54 acetylation by expressing HDAC 6 or HDAC11 could promote the deacetylation of RAD54 (Fig. 4c). Interestingly, consistent with their role in RAD54 acetylation, overexpression of GCN5 or PCAF dramatically increased the BRD9–RAD54 interaction. Meanwhile, reducing RAD54 acetylation by overexpressing HDAC 6 or 11 decreased the BRD9–RAD54 interaction (Fig. 4d). Decreased GCN5 and PCAF levels reduced RAD54 acetylation; on the other hand, knocking down HDAC 6 and 11 really increased RAD54 acetylation signal (Supplementary Fig. 8). In summary, these findings suggested that RAD54 is acetylated in response to DNA damage, and that RAD54 acetylation enhances the DNA damage-induced RAD54–BRD9 interaction.

We next investigated the acetylation site(s) of RAD54. As shown in Supplementary Figure 3A, the c terminus (450–747AA) of RAD54 was acetylated in response to DNA damage. Further mass spectrometry analysis showed that K515 was the acetylated site (Supplementary Fig. 3B, C). To confirm the functional significance of the K515 acetylation site on RAD54, we overexpressed RAD54 WT and K515R mutant RAD54 in 293T cells. As shown in Fig. 4e, the RAD54 acetylation level was increased following DNA damage in cells expressing WT RAD54 but not K515R mutant RAD54. In addition, the RAD54 K515R mutation abolished the ability of RAD54 to interact with BRD9 and induced the RAD51 chromatin persistence (Fig. 4e, Supplementary Fig. 6B). In addition, although RAD54 K515R could still bind to RAD51 at basal levels, the interaction was not enhanced by DNA damage (Fig. 4e). We further confirmed that the K515 acetylation site of BRD9 was important for BRD9 binding by synthesizing two peptides that encompass the K515 or K515-Ac sites (506–525AA) of RAD54. As shown in Supplementary Fig. 4A, BRD9 interacted with the K515-Ac peptide but not the K515 peptide. We also found that only BRD9, but not other bromodomain-containing proteins could specifically bind to K515-Ac peptide (Supplementary Fig. 4B). All these data suggest that the K515-Ac site of RAD54 is important for BRD9 binding and IR-induced RAD51–RAD54 complex formation.

We next investigated the functional significance of RAD54 acetylation. As shown in Fig. 4f, g, RAD54 K515R mutation abolished RAD54 foci formation and led to sustained RAD51 accumulation at DSBs. Moreover, compared with WT RAD54, the K515R mutation was hypersensitive to PARP inhibition and cisplatin (Fig. 4h). Taken together, these data provide strong evidence that the RAD54 K515 acetylation regulates its interaction with RAD51 at DSBs through BRD9, and that BRD9 is important for proper DDR and HR repair.

**BRD9 is overexpressed in ovarian cancer and regulates the response to DNA-damaging agents.** Many studies suggest that

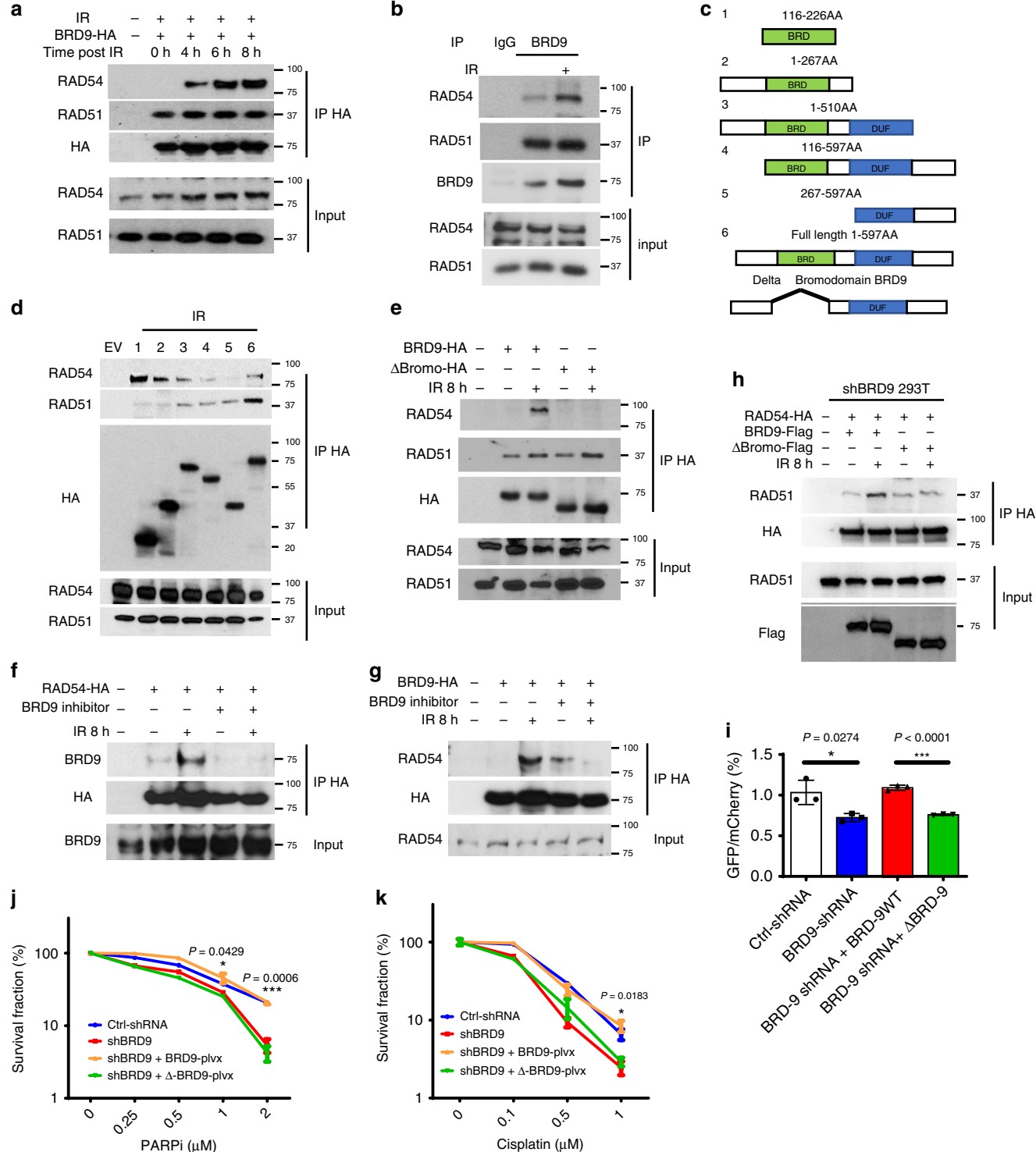

the status of the DDR pathway affects cancer cells' response to chemotherapy, with loss of DDR elements increasing the sensitivity to DNA-damaging platinum and PARP inhibition, such as BRCA1, BRCA2, BARD1, ATM, and PALB2[27]. In contrast, increased DNA repair capability is one of the factors responsible for cancer drug resistance[28]. In order to evaluate the potential role of BRD9 in therapeutic response, we assessed publicly available data from the TCGA database, and found that BRD9 is amplified in about 15% of ovarian cancers (Fig. 5a). Furthermore, overexpression of BRD9 in ovarian cancer is

significantly associated with decreased survival (Fig. 5b, HR 1.2 (95% CI: 1.06–1.37), $p = 0.0044$). Next, we assessed the expression of BRD9 in a number of ovarian cancer and normal cell lines, and found that BRD9 was overexpressed in multiple ovarian cancer cell lines (Fig. 5c). To further examine the potential role of BRD9 as a target for cancer therapy, we knocked down BRD9 in OVCAR8 and OVCAR10 cells, ovarian cancer cell lines with high expression of BRD9, and performed colony-formation assays following olaparib, cisplatin, etoposide, and IR treatment. Depletion of BRD9 significantly

**Fig. 3 BRD9 separately interacts with RAD51 and RAD54 through its C-terminal and bromodomain. a, b** BRD9 interacts with RAD54 and RAD51. **a** 293T cells were transfected with HA-tagged BRD9 plasmid. Forty-eight hours after transfection, cells were exposed to 10-Gy IR and collected at the indicated time points. Immunoprecipitation with anti-HA beads was performed. Blots were probed with the indicated antibodies. **b** OVCAR8 cells were exposed to 10-Gy IR. Lysates were collected after 8 h, and co-immunoprecipitation with anti-BRD9 antibody was performed. Blots were probed with the indicated antibodies. **c–g** BRD9 interacts with RAD51 through the C terminus, and its bromodomain mediates its interaction with RAD54. **c** Schematic diagram depicting a set of HA-tagged BRD9 expression constructs. **d** 293T cells were transfected with the indicated constructs of HA-BRD9 and exposed to 10-Gy IR. Lysates were collected after 8 h. Immunoprecipitation with anti-HA beads was performed. Blots were probed with the indicated antibodies. **e** 293T cells were transfected with the indicated constructs. Forty-eight hours later, cells were treated with 10-Gy IR 8 h before immunoprecipitation with anti-HA beads. Blots were probed with indicated antibodies. **f, g** 293T cells were transfected with the indicated constructs. Twenty-four hours later, cells were exposed to BRD9i, 10-Gy IR, or both. Lysates were collected after 8 h. Immunoprecipitation with anti-HA beads was performed. Blots were probed with the indicated antibodies. **h, k** Bromodomain of BRD9 is essential for its HR activity. **h** BRD9-knockdown 293T cells were infected with the indicated lentiviral plasmids. Twenty-four hours after transfection, cells were exposed to IR. Lysates were collected after 8 h, and immunoprecipitation with anti-HA beads was performed. Blots were probed with the indicated antibodies. **i** BRD9-knockdown 293T cells expressing the DR-GFP reporter were infected with lentivirus expressing the indicated proteins. Twenty-four hours later, cells were subjected to flow cytometry. Representative data (mean ± SEM) are shown from $n = 3$ biologically independent samples. *$p < 0.05$, ***$p < 0.001$ by two-sided unpaired $t$ test. **j, k** BRD9-knockdown OVCAR8 cells were infected with lentivirus expressing the indicated proteins, and subjected to colony-formation assay to assess the sensitivity to PARPi (**j**) and cisplatin (**k**). Representative data (mean ± SEM) are shown from $n = 3$ biologically independent samples. *$p < 0.05$, **$p < 0.01$, ***$p < 0.001$ by two-sided unpaired $t$ test.

sensitized cells to these DNA-damaging chemotherapies and radiotherapy (Fig. 5d–f, Supplementary Fig. 5A–F). On the other hand, overexpression of BRD9 in OVCAR7 cells, a cell line with low expression of BRD9, led to greater resistance to olaparib and cisplatin (Fig. 5H–J). Interestingly, the small-molecule BRD9 inhibitor, I-BRD9, sensitized BRD9-expressing cells to olaparib (Fig. 5g, Supplementary Fig. 5G), while I-BRD9 treatment did not further augment the sensitivity of non-malignant MEF cells, which harbor low expression of BRD9 to olaparib (Supplementary Fig. 5H and Supplementary Fig. 7D). These results suggest that BRD9 may be a promising and tumor-specific therapeutic target for HR-proficient cancers expressing BRD9.

In order to investigate BRD9 as a potential target for ovarian cancer therapy in vivo, we assessed the sensitivity of control and BRD9 KD OVCAR8 cells to olaparib in xenograft models. As shown in Fig. 5k, BRD9 KD did not affect tumor growth in the absence of treatment in these models. However, depletion of BRD9 significantly delayed tumor growth following olaparib treatment. In addition, compared with single-agent treatment with olaparib or I-BRD9, the combination of I-BRD9 and olaparib resulted in profound tumor growth delay in these xenograft models (Fig. 5l).

## Discussion

The BRDs are found in 42 diverse proteins in mammalian cells. These proteins recognize acetylated histones and regulate gene expression through multiple mechanisms. For example, BRD-containing proteins function as scaffolding proteins that promote the assembly of protein complexes. They also act as transcription factors or co-regulators. BRD-containing proteins are reported to be deregulated in cancer, and many cancer-causing mutations have been mapped to BRDs themselves. However, the role of BRDs in cancer remains largely unknown. By analyzing the TCGA database and performing HR repair screening, we demonstrated that three BRD-containing proteins, BRD9, ASH1L, and ZMYND8, positively regulate HR, and the mutations of these three BRD-containing proteins are associated with HR-deficient mutational signatures in ovarian cancer. ZMYND8 has been reported to identify DNA damage within transcriptionally active chromatin, and bind with CHD4 to promote HR[16]; however, the roles of BRD9 and ASH1L in HR are not clear. Interestingly, according to our screening data, depletion of ASH1L downregulated both HR and NHEJ, while depletion of BRD9 only caused HR deficiency. These results

suggest that BRD9 may specifically regulate HR pathway. BRD9 is a non-BET bromodomain-containing protein that maintains and facilitates oncogenic transcription directly by recognizing the acetylated lysine on post-translationally modified histone proteins, contributing to cancer cell proliferation and survival[29,30]. Proteomic analysis has also identified BRD9 as a dedicated member of the mammalian SWI/SNF complex[31] that is the most frequently mutated chromatin-regulatory complex in human cancer. In this study, we clarified that the bromo-domain of BRD9 binds DNA damage-induced K515 acetylation on RAD54, and promotes RAD54 binding to RAD51, which is essential for HR. Meanwhile, transcription of RAD51 and RAD54 is unaffected by the expression level or function of BRD9 (Supplementary Fig. 9D). These findings demonstrate a transcriptional regulation-independent molecular mechanism of BRD9 in HR regulation.

Recent clinical trials showed that PARP inhibitors and platinum are effective in treating ovarian cancer patients with mutations of BRCA1, BRCA2, and other genes encoding proteins involved in HR[32]. Some patients present drug resistance to these chemotherapies through HR-related factor amplification[33]. Analysis of the TCGA database showed that BRD9 is amplified in 15% of ovarian cancer patients, and higher expression of BRD9 is associated with poor outcome in ovarian cancer patients undergoing platinum-based treatment. In our study, we found that overexpression of BRD9 rendered ovarian cancer cells resistant to PARP inhibitors and platinum, while depletion of BRD9 made cells sensitive to chemotherapy, suggesting that BRD9 related ovarian cancer chemoresistance, and identifying BRD9 as a potential drug target in ovarian cancer. In this study, we further discovered that inhibition of BRD9 using I-BRD9 sensitizes ovarian cancer cells to cisplatin and PARP inhibitor in vitro and in xenograft mice models. While HR activity is not further inhibited by BRD9i in BRD9 KD cells (Supplementary Fig. 9A–C). Collectively, these results support our hypothesis that BRD9 amplification in ovarian cancers may confer higher HR activity and resistance to chemotherapy. Furthermore, the combination of PARPi/platinum and BRD9 inhibitor may help overcome chemoresistance in ovarian cancer, especially those overexpressing BRD9.

In summary, we clarified that BRD9 as a DDR factor facilitates HR through regulating RAD54–RAD51 complex, which in turn contributes to the chemoresistance of cancer cells. Considering that BRD9 is amplified in a subset of ovarian cancers, a combination of BRD9 inhibitor I-BRD9 and platinum or PARP inhibitors may provide a therapeutic approach to HR-proficient cancers.

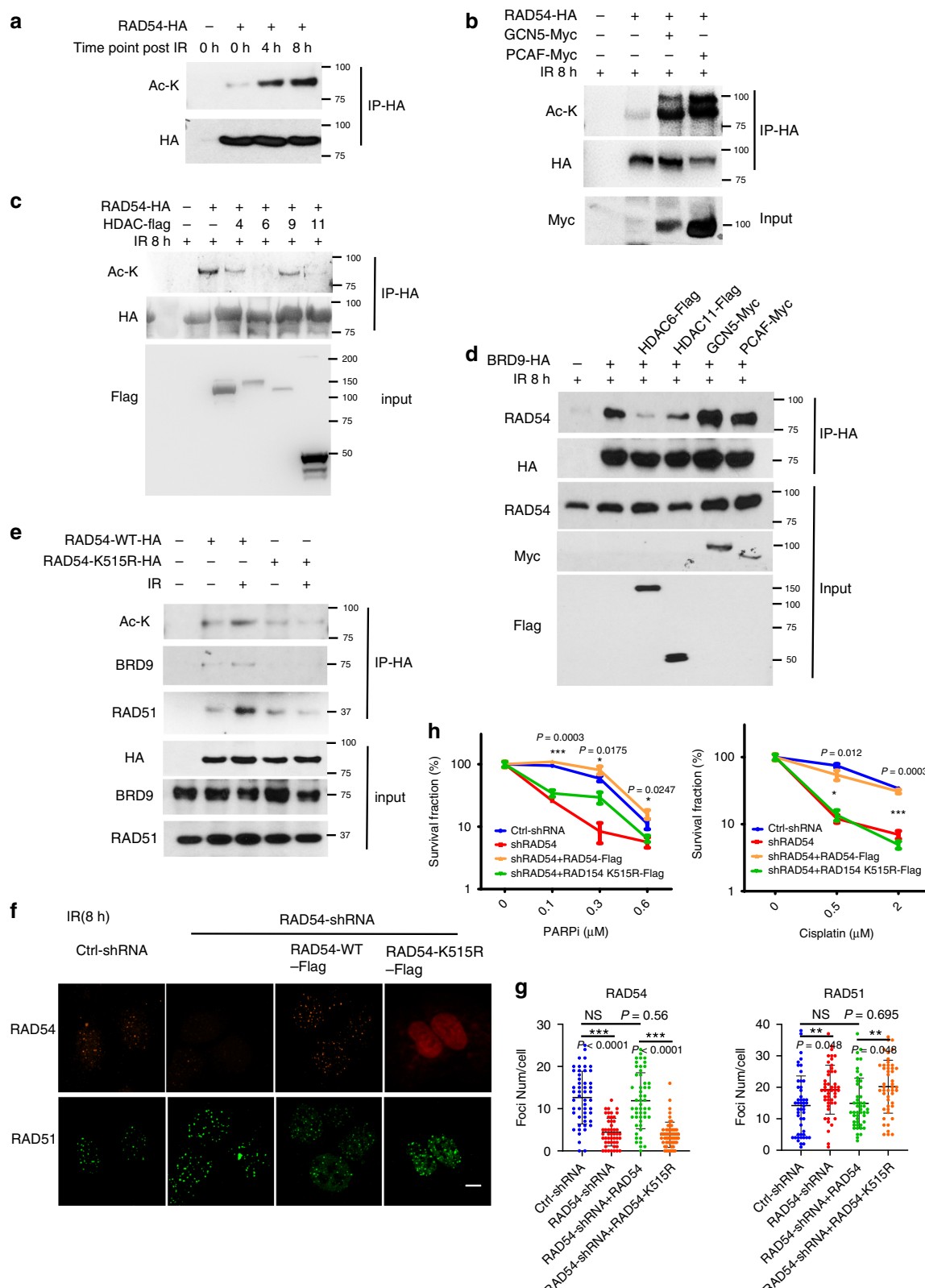

## Methods

**Cell culture and transfection**. HEK293T, Ovcar7, Ovcar8, Ovcar10, A2780, U2OS, HCT-116, MEF, MCF10A, MDA-MB-231, HCC1806, BT549, T98G, and SW480 cell lines were all purchased from ATCC. SKOV3 cells were purchased from Thermo Fisher Scientific. EFO-27 cells were purchased from DSMZ. Human ovarian epithelial cells were purchased from ScienCell. Cells were grown in DMEM RPMI1640 or McCoy's 5A with 10% FBS, and were transfected with Polyetherimide (PEI) or TransIT-X2 (Mirus).

**shRNA KD**. BRD9 shRNA (NM_023924.3), RAD54 shRNA (NM_003579), and RAD51 shRNA (NM_002875) were purchased from Sigma-Aldrich. Lentiviral BRD9 and RAD54 shRNAs were made according to the protocol from Sigma-Aldrich.

BRD9 shRNA no.1 CTCCTGGATATTCAATGATAA
BRD9 shRNA no.2 ACAAGTCAGTTACGGAATTTA
RAD54 shRNA GTGAAGATTGAGCAGGTCGTT
RAD51 shRNA GCTGAAGCTATGTTCGCCATT

**Fig. 4 RAD54 acetylation is important for BRD9 recognition and HR activity. a–c** RAD54 is acetylated by GCN5/PCAF and deacetylated by HDAC 6/ HDAC11 following induction of DNA damage. **a** 293T cells were transfected with control or RAD54-HA plasmid. Twenty-four hours after transfection, cells were exposed to the 10-Gy IR and harvested at the indicated time points. Immunoprecipitation with anti-HA beads was performed. Blots were probed with the indicated antibodies. **b** 293T cells were transfected with the indicated plasmids. Twenty-four hours after transfection, cells were exposed to 10-Gy IR, and lysates were collected after 8 h. Immunoprecipitation with anti-HA beads was performed. Blots were probed with the indicated antibodies. **c** 293T cells were transfected with the indicated plasmids, treated as indicated, and subjected to immunoprecipitation as outlined in **b**. Blots were probed with the indicated antibodies. **d, e** RAD54 K515 acetylation is important for RAD51–RAD54 complex formation. **d** 293T cells were transfected with the indicated plasmids, treated as indicated, and subjected to immunoprecipitation as outlined in **b**. Blots were probed with the indicated antibodies. **e** 293T cells were transfected with the indicated plasmids, exposed to either no IR or 10-Gy IR as indicated, and subjected to immunoprecipitation as outlined in **b**. Blots were probed with the indicated antibodies. **f–h** RAD54 K515 acetylation is essential for HR activity. **f, g** U2OS cells were transfected with the indicated plasmids and exposed to 2-Gy IR. Cells were fixed after 8 h and stained for the indicated proteins. Representative immunofluorescence images of RAD51 (green) and RAD54 (red) are shown in **f**. Quantification of the indicated foci is shown in **g**. Representative data (mean ± SEM) are shown from $n = 50$ cells examined over three independent experiments. $**p < 0.01$, $***p < 0.001$ by two-sided unpaired $t$ test. NS not significant. Scale bar, 10 μm. **h** Survival curves of U2OS cells expressing the indicated constructs and exposed to the indicated doses of PARPi or cisplatin. Representative data (mean ± SEM) are shown from $n = 3$ biologically independent samples. $*p < 0.05$, $***p < 0.001$ by two-sided unpaired $t$ test.

---

**HR/NHEJ assay**. We transfected Ovcar8 and U2OS cells with control or BRD9 shRNA, and co-transfected m-Cherry, DR-GFP/NHEJ, and I-SceI expression vectors using TransIT-X2 (Mirus). Cells were harvested 36–48 h later and analyzed by flow cytometry system Attune NxT to detect recombination events as described previously. In this experiment, m-Cherry was used to normalize for transfection efficiency.

**Plasmids and antibodies**. The RAD51 and GFP BRD9 (65380) were purchased from Addgene and subcloned into plvx3 or CMV-HA vector (635690 Clontech). pEGFP-Rad54-N1 plasmid was provided by Christel Braun lab. Rad54-GST plasmid was provided by Wolf-Dietrich Heyer lab. Both were subcloned into plvx3 or CMV-HA vector (Clontech). GCN5, PCAF, and HDAC series of plasmids were a gift from Jun Huang lab. The deletion bromodomain BRD9 and K515R RAD54 mutants were generated by a two-step mutation method using In Fusion HD cloning Plus kit from Clonetech (638909).

Anti-BRD9 antibody was purchased from Bethyl Laboratories (A303-781A). Anti-Myc (sc-40, mouse), Anti-RAD54 antibody (sc-166370), and anti-RAD51 antibody (sc-377467) for WB were purchased from Santa Cruz Biotechnology. Anti-Ack antibody was purchased from Rockland (600-401-939). Anti-γH2AX antibody (05-636) was purchased from Millipore. Anti-RAD51 antibody (ab133534) and RAD54 antibody (ab11055) for IF were purchased from Abcam. Anti-Flag (F1804), anti-HA (H9658), and anti-β-Actin (A1978) antibodies were purchased from Sigma. Anti-BRD9 (rabbit) and anti-53BP1 (rabbit) for IF was homemade; anti-SMARCC1 (11956 rabbit), anti-BRG1 (3508 rabbit), anti-BAF250 (12354 rabbit), and anti-BRM (11966 rabbit) for WB (1:1000) were purchased from Cell Signaling Technology. Anti-H3 (17168-1-AP, rabbit) for WB (1:1000) was purchased from Proteintech. Alexa Fluor® 488 AffiniPure Donkey Anti-Rabbit IgG (H + L 715-585-150), Alexa Fluor® 594 AffiniPure Donkey Anti-Rabbit IgG (H + L 711-585-152), Alexa Fluor® 488 AffiniPure Donkey Anti-Mouse IgG (H + L 715-545-150), and Alexa Fluor® 594 AffiniPure Donkey Anti-Mouse IgG (H + L 715-585-150) were purchased from Jackson Lab. Donkey Anti-Mouse IgG (H + L) ML* (715-675-151) and Donkey Anti-Rabbit IgG (H + L) ML*(711-675-152) were purchased from Jackson ImmunoResearch.

**Immunofluorescence staining**. Immunofluorescence staining was performed according to standard process. Briefly, Ovcar8 or U2OS cells were seeded and transfected with RAD54–plvx3 wild-type or K515R mutants in six-well plates containing coverslips. After treatment with the appropriate dose of irradiation, cells were fixed at the indicated time points using 3% paraformaldehyde for 30 min, washed three times in 1 × PBS, and then extracted with 0.5% Triton X-100 PBS solution for 5–10 min. After blocking with PBS containing 1% bovine serum albumin, cells were incubated with the indicated primary antibodies overnight. Cells were washed three times with 1 × PBS and incubated with Alexa Fluor 488- or Alexa Fluor 594-conjugated secondary or primary antibodies (Thermo Fisher Scientific) for 30 min–1 h at room temperature. Finally, cells were stained with 100 ng/ml 4, 6-diamidino-2-phenylindole (DAPI) for 3–5 min to visualize nuclear DNA. The coverslips were mounted onto glass slides using anti-fade solution. Finally, the slides were visualized using Leica ECLIPSE E800 fluorescence microscope with a 40× objective lens (NA 1.30).

**Co-immunoprecipitation and Western blotting**. For transient transfection and co-immunoprecipitation assays, constructs encoding different constructs were transiently co-transfected into HEK293T cells. Transfected cells were lysed with NETN buffer (20 mM Tris–HCl, pH 8.0, 1 mM EDTA, 100 mM NaCl, and 0.5% NP-40) containing protease inhibitors on ice for 30 min. Cell debris were removed by centrifugation at $13,000g$ for 10 min. Soluble fractions were gathered and incubated with the corresponding beads overnight at 4 °C. Beads were centrifuged

and washed three times with NETN buffer, boiled in 1× SDS loading buffer for 5 min, and separated on sodium dodecyl sulfate polyacrylamide gel electrophoresis (SDS-PAGE) and immunoblotted using the indicated antibodies. For endogenous immunoprecipitation assay, the cells were also lysed using NETN lysis buffer with protease and phosphatase inhibitors. The soluble fractions were collected after removing cell debris by centrifugation. About 1 mg of the whole-cell extract was then incubated with 25 μl of a 1:1 slurry of protein A–Sepharose coupled with 2 μg of the indicated antibodies for 3–4 h at 4 °C. The Sepharose beads were washed twice with NETN, boiled in 1× SDS WB loading buffer, resolved on SDS-PAGE, blocked in 5% milk TBST buffer, and then detected with antibodies as indicated. Uncropped and unprocessed scans of the most important blots now can be found in supplementary figure in the Supplementary Information.

**Colony-formation assay**. Ovcar7, Ovcar8, Ovcar10, or U2OS cells (300–3000) were seeded in triplicate in each well of six-well plates. After 8–16 h, cells were treated with indicated chemical or exposed to ionizing radiation at indicated doses, and left for 8–16 days in the incubator to allow for colony formation. Colonies were fixed with methanol, stained with GIEMSA for 10 min, and then counted. The results were normalized to the plating efficiencies of the untreated group.

**Tumor xenograft**. All experiments were performed with the approval of the Institutional Animal Care and Use Committee at Mayo Clinic (Rochester, MN). All mice used in this study were maintained under specific pathogen-free conditions, 21 ± 2 °C relative humidity of 45 ± 15%, and a 12-h light/dark cycle. Ovcar8 cells stably expressing control or BRD9 shRNA were subcutaneously injected into the flanks of 6-week-old female athymic nude Ncr nu/nu (NCI/NIH) mice using 18-gauge needles. Each mouse received two injections of a 0.2-ml mixture of 2 million cells with 50% growth factor-reduced MATRIGEL (BD Bioscience). Mice bearing tumors of 200 mm³ were randomly assigned into the indicated groups: vehicle control (saline), PARPi (purchased from Toronto Research Chemicals O514500, 50 mg kg⁻¹), iBRD9 (purchased from TOCRIS Cat. No. 5591, 40 mg kg⁻¹), and PARPi (50 mg kg⁻¹) together with iBRD9 (40 mg kg⁻¹). The treated mice were intraperitoneally injected 3 times/week. Tumor volume was measured every 7 days using calipers, and tumor volume was calculated using the formula length × width². Mice were sacrificed for tumor dissection on day 28 of treatment.

**Statistics and reproducibility**. For cell survival assay, data are presented as the mean ± S.E.M. of three independent experiments. All the Western blotting and micrograph data were repeated independently three times with similar results. For the animal xenograft study, data are presented as the mean ± S.E.M. of seven mice. Statistical analyses were performed in Microsoft Excel, GraphPad Prism7 with ANOVA, the Student's $t$ test, or $\chi^2$ test. Statistical significance is represented in figures by $*p < 0.05$; $**p < 0.01$, $***p < 0.001$, n.s., not significant. The flow cytometry data were gathered by Attune NxT Flow Cytometer software v2.6 and analyzed by flowjo.

**Patient HR signature screening**. We selected 75% as the threshold for high HR mutation signatures. An ovarian cancer patient will be considered to be a patient with high HR mutation signature if the quantification of HR mutation signature of this patient is higher than that of 75% of the total patients with ovarian cancer. When the BRD proteins are mutated, the percentage of patients with high HR mutation signature is significantly higher than when BRD proteins are not mutated. In other words, the mutation of these BRD proteins is closely associated with upregulation of HR mutation signature, suggesting that their mutation might cause HR deficiency. The clinical data for ovarian cancer patients from the TCGA

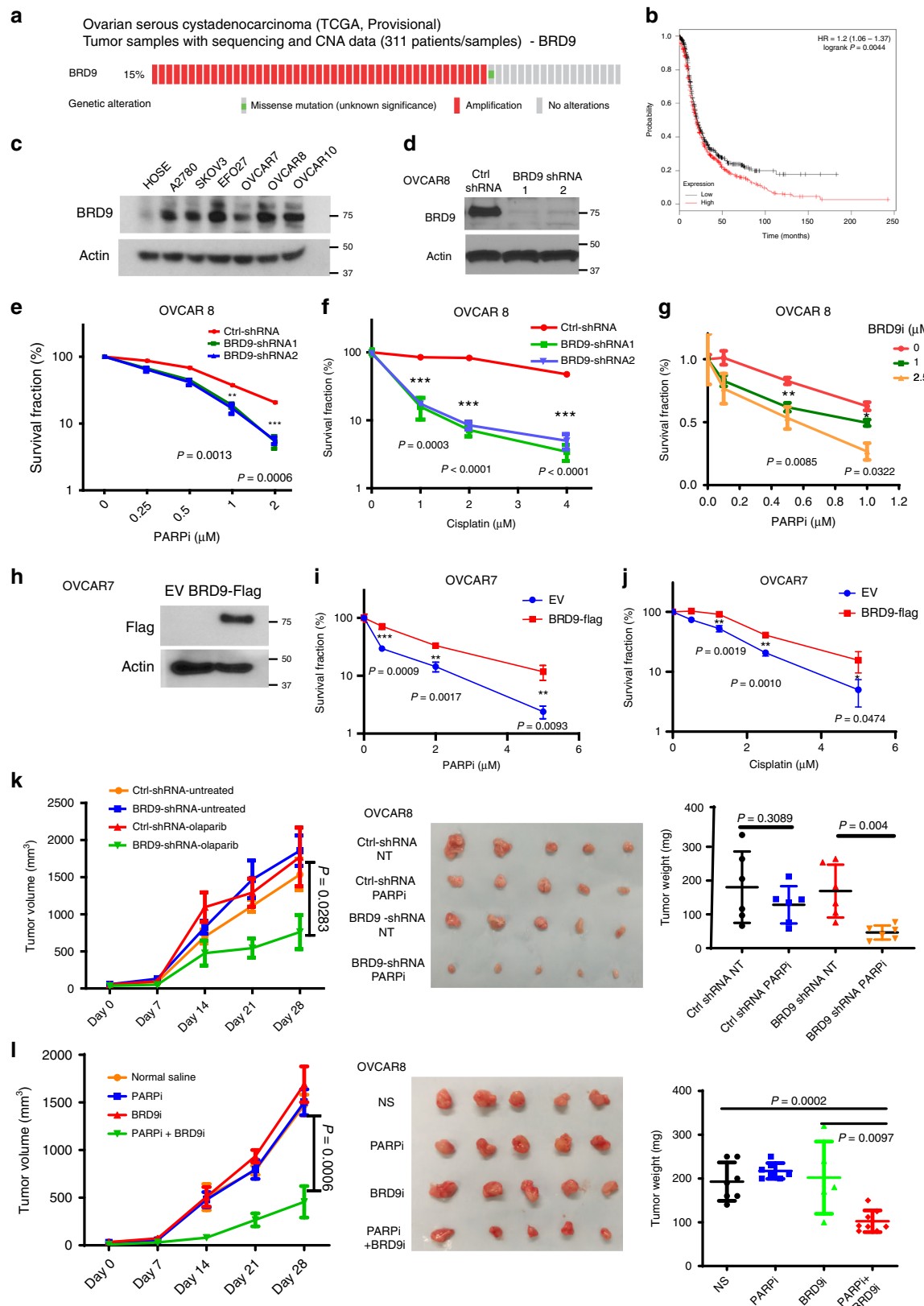

consortium were obtained from the Broad Institute Genomic DNA Affinity Chromatography Firehose (https://gdac.broadinstitute.org/).

**Reporting summary**. Further information on research design is available in the Nature Research Reporting Summary linked to this article.

## Data availability

All the data are available from the corresponding authors upon reasonable request. The data underlying Figs 1a–f, h, j, 2b–g, 3a, b, d–k, 4a–e, g, h, 5c–l and Supplementary Figs 1A–G, 2A, B, D, E, 3A, E, 4A, B, 5B–H, 6A, B, 7A–C, 8A–D, 9B–D can be found in the Source Data file. A reporting summary for this article is available as a Supplementary Information file.

**Fig. 5 BRD9 is overexpressed in ovarian cancer and regulates response to DNA-damaging agents. a–c** BRD9 is overexpressed in ovarian cancer patient samples and cell lines, and is associated with decreased patient survival. **a** Analysis of BRD9 expression in ovarian cancer patient samples from the TCGA database. Red indicates BRD9 amplification. **b** Kaplan–Meier curve showing the survival of ovarian cancer patients with low ($n = 720$) and high ($n = 715$) expression of BRD9. Statistical analysis with the two-sided log-rank (Mantel–Cox) test revealed statistically significant differences as shown on the graph. **c** Immunoblot of indicated proteins in the indicated ovarian cancer cell lines. **d–g** Depletion/inhibition of BRD9 sensitizes OVCAR8 cells to PARPi and cisplatin. **d–f** OVCAR8 cells infected with indicated shRNAs and subjected to WB assay (**d**) to assess knockdown efficiency, and colony-formation assay to assess the sensitivity to olaparib (**e**) and cisplatin (**f**). **g** OVCAR8 cells were exposed to olaparib and/or BRD9i and subjected to colony-formation assay. Representative data (mean ± SEM) are shown from $n = 3$ biologically independent samples. **p < 0.01,***p < 0.001 by two-sided unpaired $t$ test. **h–j** Overexpression of BRD9 confers resistance to PARPi/cisplatin. OVCAR7 cells infected with the indicated lentivirus and subjected to colony-formation assay to assess the sensitivity to olaparib (**i**) and cisplatin (**j**). Protein expression was assessed by Western blot (**h**). Representative data (mean ± SEM) are shown from $n = 3$ biologically independent samples. **p < 0.01,***p < 0.001 by two-sided unpaired $t$ test. **k, l** Depletion/inhibition of BRD9 sensitizes OVCAR8 cells to PARPi in vivo. **k** OVCAR8 cells expressing the indicated shRNAs were subcutaneously injected into the flank of NOD-SCID mice. Mice were treated with or without olaparib (50 mg/kg i.p. 3 days × 8 times). Tumor volume and weight were assessed (**l**). OVCAR8 cells were subcutaneously injected into the flank of NOD-SCID mice. Mice were treated with vehicle (normal saline, NS), olaparib (PARPi, 50 mg/kg), BRD9i (40 mg/kg i.p. 3 days × 8 times), or olaparib + BRD9i. Tumor volume and weight were assessed. Representative data (mean ± SEM) are shown from $n = 6$ biologically independent samples by two-sided unpaired $t$ test.

## Code availability

The code used for HR signature screening is available at https://github.com/chaohz/My_BRD9_HR_mutation_analysis.

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

## Acknowledgements

We thank Wolf-Dietrich Heyer, Christel Braun, and Jun Huang labs for providing plasmids. Thank you to the members of Lou lab for helpful discussions. We would also like to thank the Mayo Clinic Cancer Center for the use of the core facilities. This work was supported by the National Natural Science Foundation of China (91749115, 81572770, 81872298, and 81802754), the Natural Science Foundation of Jiangxi Province (20181ACB20021), and Mayo foundation.

## Author contributions

Q.Z., J.H., C.Z., F.Z., and Y.Z. designed and performed the experiments. Qian Z., Y. Chen, and X.T. assisted with the in vivo experiments. J.H. and M.D. assisted with microscopy. Q.Z., W.K., S.N., B.Q., B.L., and K.L. analyzed the data and wrote the paper. J.Y., R.W.M., and Z.L. conceived and supervised the project.

## Competing interests

The authors declare no competing interests.
