## [Peer Review File · Nature Communications]

Reviewers' comments:

Reviewer #1 (Remarks to the Author):

In this manuscript, the authors report that BRD9 participates in HR repair by facilitating the removal of RAD51 from the sites of DNA damage. The authors found that RAD51 mediated the recruitment of BRD9 to DNA lesions. The bromodomain of BRD9 recognized acetylated K515 of RAD54, and mediated the recruitment of RAD54 to the sites of DNA damage, which replaced RAD51 for the completion of HR repair. Moreover, BRD9 is overexpressed in ovarian cancer, and inhibition of BRD9 sensitized ovarian cancer cells to PARP inhibitor treatment. Overall, the authors reveal a novel molecular mechanism in DNA damage repair field, which may impact for the clinical treatment for ovarian cancer patients. The specific points listed below should be addressed before publication.:

1. The figure legend should be revised by including relatively detailed information, so that readers may easily understand the results of each assay.
2. The authors need to provide the evidence or discuss the specificity of the acetylated K515 that is recognized by the bromodomain of BRD9, since other members in the BRD family may not bind RAD54.
3. The author only examined GCN5 and PCAF. However, other acetyltransferases are known to play key roles in DNA damage repair, such as TIP60, p300 and MOF. The authors need to examine the possibilities that other enzymes may be also involved in the acetylation of RAD54.
4. The retention of RAD51 should be examined in cells only expressing the K515A mutant of RAD54 and the bromodomain deletion mutant of BRD9.

Reviewer #2 (Remarks to the Author):

The authors of this manuscript made the interesting observation that mutations of several BRD proteins are associated with the mutation signature 3 in cancer, indicating that these BRD proteins may be involved in HR. Their subsequent experiments confirmed that BRD9 is important for HR. Interestingly, ablation of BRD9 did not alter RAD51 foci but reduced RAD54 foci. Moreover, BRD9 binds RAD51 and is required for the damage induced binding between RAD51 and RAD54. Through a set of careful biochemistry experiments, the authors found that the bromo domain of BRD9 is important for binding RAD54, and the K515 of RAD54 is acetylated after damage and required for BRD9 binding. Deletion or inhibition of the bromo domain of BRD9, or mutation of the K515 of RAD54, increased the sensitivity of cells to PARPi and cisplatin. Loss or inhibition of BRD9 also increased the chemo sensitivity of tumors in mouse xenografts. These results strongly suggest that BRD9 is a regulator of HR that affects chemo sensitivity of tumors. The results of this study (especially the biochemistry data) are compelling, and the conclusion is important. I have a few suggestions to the authors to strengthen the main conclusions. A satisfactorily revised manuscript should be suitable for publication in Nature Communications.

1. In Fig. 2A, merged images of RAD51 and RAD54 foci should be shown. The colocalization of RAD51 and RAD54 looks quite partial from the images.
2. The IF data in Fig. 2G and S2E should be quantified.
3. In Fig. S2C, BRD9i seems to inhibit RAD51 foci formation. Why?
4. In Fig. S3A, is the acetylation of endogenous RAD54 at K515 detectable by mass spec after IR?
5. In Fig. 4, can the authors show that BRD9 binds to an acetylated peptide encompassing the K515 of RAD54? The K515R mutant may be defective for BRD9 binding because of a change in the binding surface but not the lack of acetylation. Direct evidence for K515-Ac mediated interaction is

important.

6. In Fig. 4, does knockdown of GCN5, PCAF, DAC6, or HDAC11 affect RAD54 acetylation? The specificity of these enzymes to RAD54 would be strengthened by depletion experiments.

7. In Fig. 5C, the differences between BRD9 levels in cancer cell lines seem to quite small. For example, the BRD9 levels in OVCAR8 and 10 are only 2-3 folds higher than OVCAR7. If BRD9 is only reduced 2-3 folds in OVCAR8/10 or only increased 2-3 folds in OVCAR7, does it significantly increase or decrease PARPi/cisplatin sensitivities?

8. The xenograft data in Fig. 5 are impressive. Is BRD9 expressed in normal cells? Does BRD9 depletion or inhibition cause any deleterious effects in normal cells? Does depletion or inhibition of BRD9 sensitize normal cells to PARPi too?

Reviewer #3 (Remarks to the Author):

In the current study entitled "Bromodomain Containing Protein BRD-9 Orchestrate the RAD51-RAD54 Complex and Coordinately Regulate Homologous Recombination" Zhou and colleagues examine the potential role of BRD9 in DNA repair. Studying the underlying mechanisms associated with DNA repair remains pertinent, given their fundamental importance in many aspects of biology. Moreover, these mechanisms frequently become perturbed in diseases including multiple cancers; likely contributing to disease development and also presenting therapeutic vulnerabilities. In this study the authors suggest that BRD9 plays an important role in DNA repair through interactions with the key DNA repair proteins RAD51 and RAD54. Based on these reports, the authors further suggest that targeting BRD9 function may provide an opportunity to achieve improved therapeutic responses in tumours treated with PARP inhibitors. Although the findings of this study are interesting, there are currently a number of shortcomings with the experimentation that should be addressed in order to strengthen the findings.

Major Points:

1. The authors begin by examining mutational profiles in published TCGA cancer genomics data. They suggest that BRD9 together with some other bromodomain containing proteins (ASH1L, BRWD3, KMT2A, ZMYND8 and SMARCA4) are mutated with greater frequency in Signature 3 ovarian cancers. It is important to note here that the multiple distinct cancer mutational signatures that have been described have different mutational frequencies/preferences. As such it remains important to delineate whether the observed mutational rate for these genes is simply an indirect consequence of their underlying DNA sequence. How or whether the underlying DNA sequence of these genes differ from other bromodomain containing genes, and other genes in general, may explain the observed differences. Moreover, it could strengthen (or rule out) the suggestion that mutation of these genes may be causally linked to oncogenesis in these tumours.

2. The experimentation used to suggest that BRD9 is specifically important (compared to most other bromodomain containing proteins) for homologous recombination (HR) are weak. For example, none of the shRNA sequences used in Figure 1B are validated with regard to their knockdown efficiency. It is impossible to conclude that BRD9 is more important than many other bromodomain containing proteins in these assays without providing evidence that all shRNAs used in these assays work effectively. Moreover, in relation to these assays no positive control shRNAs (ie. those knocking down expression of known HR regulators) are used to support the robustness of the data presented in Fig. 1a.

3. The authors suggest that BRD9 bromodomain inhibition using a published small-molecule inhibitor reduces HR levels in cells. It is important to note here that this, and other BRD9 targeting small-molecules have been reported to induce broad transcriptional changes. As such it's hard to

conclude that the effects observed here are a direct consequence of bromodomain inhibition and not related to altered global transcriptional dynamics. Moreover, the doses used (10uM and 20uM) are extremely high and therefore potentially leading to additional off-target effects.

4. The immunofluorescence (IF) images and experimentation presented in Figures 1/S1, 2/S2 have no positive controls (ie. knockdown of known regulators of HR) and in several instances only use a single BRD9 shRNAs. These experiments need to be strengthened significantly to support the authors suggestions.

5. The resolution of the images presented in Fig. S2A is very poor making it nearly impossible to see the underlying numbers. Moreover, the cell cycle profiles themselves do not match the authors assertion that there is no significant effect on cell cycle profile following BRD9 knockdown. In fact, it is quite clear to see that there are proportionally fewer cells in G2/M (and possibly S) following BRD9 depletion. This is clearly an important point given that these DNA double-strand break repair mechanisms preferentially occur at certain points throughout the cell cycle. Shifting cell cycle dynamics as appears to be the case here could alter these mechanisms and their regulation indirectly.

6. In Fig. S2B the authors present Western blot data that they claim is representative of the subcellular "chromatin" fraction. However, it cannot be concluded based on the presented data that this is in fact the case. The authors have not run cytoplasmic and/or nucleoplasmic fractions on these blots to demonstrate that they have in fact adequately fractionated protein samples. This should be done and control Western blots for proteins present in non-overlapping fractions should be included to support the validity of the claims.

7. In Fig. 4 the authors over-express GCN5 and PCAF in cells and demonstrate that acetylation levels of RAD54 increases in this setting. However, no conceptual rationale was presented as to why these particular enzymes were chosen. Are these enzymes actually the primary (and biologically relevant) mediators of RAD54 acetylation? Its hard to conclude based on the presented data (and lack of rationale) whether this is the case.

Minor Points:

1. The grammar throughout the manuscript is poor and should be improved to make the text easier to read/follow.

2. The authors focus their computational analysis of cancer mutational signatures in Figure 1 exclusively on ovarian cancer. Many other cancers, in particular BRCA1/2 mutated breast cancers, have overlapping mutational signatures (ie. Signature 3). It would be interesting to note whether or not the observations made here in ovarian cancer carry over to other Signature 3 tumours; or whether they are in fact specific to ovarian cancer (also see Major Point 1).

Reviewers' comments:

Reviewer #1 (Remarks to the Author):

In this manuscript, the authors report that BRD9 participates in HR repair by facilitating the removal of RAD51 from the sites of DNA damage. The authors found that RAD51 mediated the recruitment of BRD9 to DNA lesions. The bromodomain of BRD9 recognized acetylated K515 of RAD54, and mediated the recruitment of RAD54 to the sites of DNA damage, which replaced RAD51 for the completion of HR repair. Moreover, BRD9 is overexpressed in ovarian cancer, and inhibition of BRD9 sensitized ovarian cancer cells to PARP inhibitor treatment. Overall, the authors reveal a novel molecular mechanism in DNA damage repair field, which may impact for the clinical treatment for ovarian cancer patients. The specific points listed below should be addressed before publication.:

1. The figure legend should be revised by including relatively detailed information, so that readers may easily understand the results of each assay.

Response: Thank you very much for your suggestion. We have added additional details to the figure legends to improve readability and facilitate comprehension of the results.

2. The authors need to provide the evidence or discuss the specificity of the acetylated K515 that is recognized by the bromodomain of BRD9, since other members in the BRD family may not bind RAD54.

Response: Thank you for this suggestion. We have addressed this comment by screening other bromodomain containing proteins such as BAZ1B, BAZ2A, CECR2, TRIM66, SMARCA4, BRD7 and BRD9 for their ability to interact with RAD54 K515 peptide. As displayed in **Fig.S4B**, we found that none of these other bromodomain containing proteins could interact with RAD54 K515, suggesting that the K515 RAD54 acetylation site may be specific for binding to the bromodomain of BRD9.

3. The author only examined GCN5 and PCAF. However, other acetyltransferases are known to play key roles in DNA damage repair, such as TIP60, p300 and MOF. The authors need to examine the possibilities that other enzymes may be also involved in the acetylation of RAD54.

Response: Thank you very much for your suggestion. As requested, we screened whether other key acetyltransferase MOF, TIP-60 and P300 can acetylate RAD54. As displayed in **Figure.S3E** only GCN5 and PCAF, and not MOF, TIP60 or P300 acetylate RAD54.

4. The retention of RAD51 should be examined in cells only expressing the K515A mutant of RAD54 and the bromodomain deletion mutant of BRD9.

Response: We appreciate the suggestion. As requested, we have further examined the retention of RAD51 in cells expressing K515R mutant of RAD54 or bromodomain deletion mutant of BRD9. As shown in **Fig.S6A-B**, RAD51 retention can be rescued by RAD54-WT but not RAD54-K515R, and BRD9-WT but not BRD9 bromodomain deletion.

Reviewer #2 (Remarks to the Author):

The authors of this manuscript made the interesting observation that mutations of several BRD proteins are associated with the mutation signature 3 in cancer, indicating that these BRD proteins may be involved in HR. Their subsequent experiments confirmed that BRD9 is important for HR. Interestingly, ablation of BRD9 did not alter RAD51 foci but reduced RAD54 foci. Moreover, BRD9 binds RAD51 and is required for the damage induced binding between RAD51 and RAD54. Through a set of careful biochemistry experiments, the authors found that the bromo domain of BRD9 is important for binding RAD54, and the K515 of RAD54 is acetylated after damage and required for BRD4 binding. Deletion or inhibition of the bromo domain of BRD9, or mutation of the K515 of RAD54, increased the sensitivity of cells to PARPi and cisplatin. Loss or inhibition of BRD9 also increased the chemo sensitivity of tumors in mouse xenografts. These results strongly suggest that BRD9 is a regulator of HR that affects chemo sensitivity of tumors. The results of this study (especially the biochemistry data) are compelling, and the conclusion is important. I have a few suggestions to the authors to strengthen the main conclusions. A satisfactorily revised manuscript should be suitable for publication in Nature Communications.

1. In Fig. 2A, merged images of RAD51 and RAD54 foci should be shown. The colocalization of RAD51 and RAD54 looks quite partial from the images.

Response: Thank you for this suggestion. As requested, the merged images of RAD51 and RAD54 foci are now shown. As displayed in figure 2A, RAD51 and RAD54 co-localized following IR. For the WT OVCAR8 cells, about 60% RAD51 can co-localize with RAD54 foci, which is consistent with a previous publication (BLM helicase stimulates the ATPase and chromatin remodeling activities of RAD54, Vivek Sivastava et al. 2009. J Cell Sci.). For the BRD9 KD cells, only 5% of RAD51 colocalized with RAD54 foci, suggesting BRD9 plays an important role in RAD51-RAD54 colocalization.

2. The IF data in Fig. 2G and S2E should be quantified.

Response: As requested, we have quantified the IF foci data in Fig. 2G and S2E.

3. In Fig. S2C, BRD9i seems to inhibit RAD51 foci formation. Why?

Response: We quantified the RAD51 foci in Fig S2C. BRD9i did not inhibit and instead increased RAD51 foci formation. We selected high resolution images to represent the data.

4. In Fig. S3A, is the acetylation of endogenous RAD54 at K515 detectable by mass spec after IR?

Response: As suggested, we immunoprecipitated endogenous RAD54 and performed mass spectrometry analysis. However, the K515 acetylation of endogenous RAD54 were not detected by mass spectrometry. It may be due to technical limitations including the quality and quantity of RAD54 IP sample. Exogenous RAD54 K515 acetylation can be readily detected by mass spectrometry (Fig S3).

5. In Fig. 4, can the authors show that BRD9 binds to an acetylated peptide encompassing the K515 of RAD54? The K515R mutant may be defective for BRD9 binding because of a change in

the binding surface but not the lack of acetylation. Direct evidence for K515-Ac mediated interaction is important.

Response: Thank you for this comment. We synthesized two peptides which encompass the K515 or K515-Ac site of RAD54 and performed pull down assay. As shown in **Figure S4A**, BRD9 binds with the K515-Ac peptide but not K515 peptide, suggesting that K515-Ac of RAD54 is important for BRD9 binding.

6. In Fig. 4, does knockdown of GCN5, PCAF, DAC6, or HDAC11 affect RAD54 acetylation? The specificity of these enzymes to RAD54 would be strengthened by depletion experiments.

Response: We utilized shRNA to knockdown GCN5, PCAF, HDAC6 and HDAC11, then transfected RAD54 to detect its acetylation. Decreased GCN5 and PCAF level reduced RAD54 acetylation. On the other hand, knocking down HDAC6 and HDAC11 increased RAD54 acetylation signal. This data is now shown in **FigS8**.

7. In Fig. 5C, the differences between BRD9 levels in cancer cell lines seem to quite small. For example, the BRD9 levels in OVCAR8 and 10 are only 2-3 folds higher than OVCAR7. If BRD9 is only reduced 2-3 folds in OVCAR8/10 or only increased 2-3 folds in OVCAR7, does it significantly increase or decrease PARPi/cisplatin sensitivities?

Response: Thank you for this question. For further clarification, we quantified the WB signal in **Fig.5A and B**. Data showed that BRD9 expression in OVCAR8/10 are 6-7 folds higher than OVCAR7, while BRD9 KD OVCAR8 express the protein at 19-20 folds lower level than WT OVCAR8 cells. We also tested endogenous and exogenous BRD9 in BRD9 overexpressed OVCAR7 cells. Our data showed that exogenous BRD9 expression in OVCAR7 cells is dramatically higher than endogenous to confer resistance to PARPi and Cisplatin (**Fig5 I and J**).

8. The xenograft data in Fig. 5 are impressive. Is BRD9 expressed in normal cells? Does BRD9 depletion or inhibition cause any deleterious effects in normal cells? Does depletion or inhibition of BRD9 sensitize normal cells to PARPi too?

Response: Thank you for this comment. We have assessed the expression of BRD9 in several cell lines and found that the BRD9 expression level is lower in normal cell lines such as MEF and MCF10A compared to cancer cell lines, potentially providing a therapeutic window (**Fig.S7C**). Indeed, BRD9i did not increase PARPi sensitivity in normal cells (**Fig.S4H**). Similar findings were also observed in HOSE cells (**Fig.S7D**). These data suggest that inhibition of BRD9 does not further lead to PARPi sensitivity in normal cells.

Reviewer #3 (Remarks to the Author):

In the current study entitled “Bromodomain Containing Protein BRD-9 Orchestrate the RAD51-RAD54 Complex and Coordinately Regulate Homologous Recombination” Zhou and colleagues examine the potential role of BRD9 in DNA repair. Studying the underlying mechanisms associated with DNA repair remains pertinent, given their fundamental importance in many aspects of biology. Moreover, these mechanisms frequently become perturbed in diseases including multiple cancers; likely contributing to disease development and also presenting therapeutic vulnerabilities. In this study the authors suggest that BRD9 plays an important role in DNA repair through interactions with the key DNA repair proteins RAD51 and RAD54. Based on these reports, the authors further suggest that targeting BRD9 function may provide an opportunity to achieve improved therapeutic responses in tumours treated with PARP inhibitors. Although the findings of this study are interesting, there are currently a number of shortcomings with the experimentation that should be addressed in order to strengthen the findings.

Major Points:

1. The authors begin by examining mutational profiles in published TCGA cancer genomics data. They suggest that BRD9 together with some other bromodomain containing proteins (ASH1L, BRWD3, KMT2A, ZMYND8 and SMARCA4) are mutated with greater frequency in Signature 3 ovarian cancers. It is important to note here that the multiple distinct cancer mutational signatures that have been described have different mutational frequencies/preferences. As such it remains important to delineate whether the observed mutational rate for these genes is simply an indirect consequence of their underlying DNA sequence. How or whether the underlying DNA sequence of these genes differ from other bromodomain containing genes, and other genes in general, may explain the observed

differences. Moreover, it could strengthen (or rule out) the suggestion that mutation of these genes may be causally linked to oncogenesis in these tumours.

Response:

Thank you for this comment. In our initial screen we identified mutations in 6 bromodomain containing genes (BRD9, ASH1L, BRWD3, KMT2A, ZMYND8 and SMARCA4) in ovarian cancer to be associated with signature 3 which is known as a homologous recombination deficiency gene signature. We cannot totally rule out the possibility that mutation in these genes could be an indirect consequence of their underlying DNA sequence (i.e. passenger mutations in the context of repair deficiency of a different etiology). However, we still prefer our explanation because of the following reasons.

1. If the Signature 3 has the mutational preferences on the DNA sequence of these genes (BRD9, ASH1L, BRWD3, KMT2A, ZYMND8 and SMARCA4), it might be also found in other Signature 3 cancer types, i.e. BRCA. However, we did not find that. (Refer to the response to your minor points 2);

2. As we know, during DNA replication, HR repair is the only known error-free repair way for cells to guarantee high-fidelity of genome transmission. If there are sequence preferences for HR repair on these bromodomain-containing proteins, it means these genes should be mutated with greater frequency in Signature 3 ovarian cancers. We also didn't found that in ovarian cancer patient database(430 patient in total). And it seems to be not reasonable for the lack of error-free repair way for some specific genes with specific potential sequence in cells.

Threshold for high HR mutation signature: 75%				
	normal	mutation	p value	Mutation
BRD4	0.247663551	0.375	0.3207	8
TAF1L	0.237980769	0.5	0.01197	20
ATAD2	0.242424242	0.714285714	0.01222	7
BRD7	0.250574713	0	1	1
KAT2A	0.248847926	0.5	0.4379	2
TRIM24	0.25	0.25	0.6851	4
ATAD2B	0.247113164	0.666666667	0.1556	3
BRD8	0.241784038	0.6	0.01851	10
KAT2B		no mutation		
TRIM66		no mutation		
BAZ1A	0.244186047	0.666666667	0.03654	6
BRD9	0.237647059	0.727272727	0.00102	11
KMT2A	0.218362283	0.636363636	1.08E-06	33
ZMYND11	0.250574713	0	1	1
BAZ1B	0.243559719	0.555555556	0.04713	9
BRDT	0.247685185	0.5	0.2612	4
PHIP	0.243559719	0.555555556	0.04713	9
ZMYND8	0.24	0.636363636	0.006857	11
BAZ2A	0.248847926	0.5	0.4379	2
BRPF3	0.246511628	0.571428571	0.06896	7
SMARCA2	0.243559719	0.555555556	0.04713	9
BPTF	0.248826291	0.3	0.4753	10
BRWD1	0.243498818	0.461538462	0.0771	13
SP100	0.247058824	0.363636364	0.2851	11
BRD1	0.244755245	0.428571429	0.06896	7
BRWD3	0.236533958	0.888888889	8.90E-05	9
SP110	0.249422633	0.333333333	0.5791	3
BRD2	0.252314815	0	1	4
CECR2	0.248837209	0.333333333	0.4667	6
SP140	0.245901639	0.444444444	0.1636	9
ASH1L	0.237529691	0.6	0.003583	15
EP300	0.238207547	0.666666667	0.002421	12
BRD3	0.251152074	0	1	2
CREEBBP		no mutation		
SP140L	0.248847926	0.5	0.4379	2
TRIM33	0.249422633	0.333333333	0.5791	3
BAZ2B	0.257211538	0.2	0.7592	10
TRIM27	0.251162791	0.166666667	0.8241	6
TAF1L	0.237980769	0.5	0.01197	20
PBRM1	0.24537037	0.75	0.04997	4
SMARCA4	0.238095238	0.5625	0.006474	16

3. Moreover, BRD9 mutations were associated with functional HR deficiency by the well established DR GFP assay(Fig.1B-C). ZYMND8 was also reportedly involved in DSB repair in recent studies. Therefore, it is highly likely that loss of BRD9 was the source of the HR deficiency and signature 3 in these tumors..

2. The experimentation used to suggest that BRD9 is specifically important (compared to most other bromodomain containing proteins) for homologous recombination (HR) are weak. For example, none of the shRNA sequences used in Figure 1B are validated with regard to their knockdown efficiency. It is impossible to conclude that BRD9 is more important than many other bromodomain containing proteins in these assays without providing evidence that all shRNAs used in these assays work effectively. Moreover, in relation to these assays no positive control shRNAs (ie. those knocking down expression of known HR regulators) are used to support the robustness of the data presented in Fig. 1a.

Response: Thank you for your comment. We have confirmed the shRNA knockdown efficiency of a majority of the bromodomain containing proteins using RT-PCR(See the following table). We also employed RAD51shRNA and 53BP1 shRNA as positive control, respectively, in Fig.1B.

Table		
	mRNA expression compare to control shRNA	
	$2^{(-\Delta\Delta CT)}$	$\Delta\Delta CT$
ASH1L	0.408419463	1.291876
ATAD2	0.501693447	0.995122
ATAD2B	0.292128294	1.775326
BAZ1A	0.377391879	1.405865
BAZ1B	0.459349831	1.122335
BAZ2A	0.254029333	1.976933
BAZ2B	0.388035083	1.365741
BPTF	0.270512634	1.886232
BRD1	0.250750221	1.995677
BRD2	0.442509919	1.176218
BRD3	0.336123361	1.572937
BRD4	0.303678216	1.719385
BRD7	0.168662109	2.567792
BRD8	0.470478824	1.087798
BRD9	0.442165092	1.177343
BRDT	0.508155477	0.976658
BRPF1	0.272071382	1.877943
BRPF3	0.218478397	2.194437
BRWD1	0.203371303	2.297812
BRWD3	0.413518811	1.273975
CBP	0.288278566	1.794465

CECR2	0.312318	1.678912
EP300	0.317553739	1.654927
KAT2A	0.365449699	1.452255
KAT2B	0.43340288	1.206219
KMT2A	0.420485001	1.249874
PBRM1	0.54360056	0.879381
PHIP	0.507150084	0.979515
SMARCA4	0.29386093	1.766795
SP110	0.199475171	2.325719
SP140	0.337444042	1.56728
SP140L	0.390750653	1.35568
TAF1	0.337301061	1.567891
TAF1L	0.330404733	1.597694
TRIM24	0.309197991	1.693397
TRIM28	0.52349648	0.933748
TRIM33	0.441495712	1.179529
TRIM66	0.347745174	1.523898
ZYMND8	0.426949175	1.227864
ZYMND11	0.21053585	2.247862

3. The authors suggest that BRD9 bromodomain inhibition using a published small-molecule inhibitor reduces HR levels in cells. It is important to note here that and other BRD9 targeting small-molecules have been reported to induce broad transcriptional changes. As such it's hard to conclude that the effects observed here are a direct consequence of bromodomain inhibition and not related to altered global transcriptional dynamics. Moreover, the doses used (10uM and 20uM) are extremely high and therefore potentially leading to additional off-target effects.

Response: To address the reviewer's question, we depleted BRD9 in cells using shRNA and further treated the cells with BRD9 inhibitor (20µM), and examined the RAD51/RAD54 foci formation and HR. As shown in **Fig. S9A-B**, BRD9 inhibitor did not further affect RAD51/RAD54 foci formation and HR in BRD9 cells, suggesting there is no off-target effect of BRD9i in HR regulation. We analyzed the RNA-seq data from previous publication and found that both BRD9-knockdown and BRD9i treatment did not affect the expression of DDR gene (*Sensitivity and engineered resistance of myeloid leukemia cells to BRD 9 inhibition, Anja Hohmann et al. 2016. Nature Chem bio.*). We also examined RAD51/RAD54 transcription level in these cells. As shown in **Fig. S9C-D**, neither depletion of BRD9 nor inhibiting BRD9 affected the transcription of RAD51/RAD54, suggesting BRD9 inhibitor does not reduce HR through regulating the transcription of RAD51/RAD54.

DMSO compare to BRD9i treated

test_id	gene_id	gene	locus	sample_1	sample_2	status	value_1	value_2	log2(fold)	test_stat	p_value	q_value	significant
Atm	Atm	Atm	chr9:5324:	296351_1	296351_3	OK	5.86017	5.75311	-0.0266	0.155225	0.876644	0.999381	no
Atr	Atr	Atr	chr9:9575:	296351_1	296351_3	OK	8.27878	7.34281	-0.17309	1.01969	0.307874	0.908763	no
Atrip	Atrip	Atrip	chr9:1089:	296351_1	296351_3	OK	8.85243	8.24431	-0.10267	0.60287	0.546595	0.983362	no
Bard1	Bard1	Bard1	chr1:7107:	296351_1	296351_3	OK	7.33403	7.65384	0.061577	-0.37042	0.711073	0.998231	no
Brc1	Brc1	Brc1	chr11:101:	296351_1	296351_3	OK	11.6448	12.7038	0.125569	-0.7215	0.470602	0.963743	no
Brc2	Brc2	Brc2	chr5:1513:	296351_1	296351_3	OK	4.21988	4.0355	-0.06446	0.503988	0.61427	0.987539	no
Brd9	Brd9	Brd9	chr13:740:	296351_1	296351_3	OK	23.3349	23.2371	-0.00606	0.036121	0.971186	0.999381	no
Chek1	Chek1	Chek1	chr9:3651:	296351_1	296351_3	OK	10.3997	9.77502	-0.08937	0.537431	0.59097	0.986887	no
Chek2	Chek2	Chek2	chr5:1112:	296351_1	296351_3	OK	10.2077	11.2302	0.137725	-0.81342	0.415978	0.949004	no
Dclre1c	Dclre1c	Dclre1c	chr2:3341:	296351_1	296351_3	OK	8.87355	8.84032	-0.00541	0.037297	0.970248	0.999381	no
Dna2	Dna2	Dna2	chr10:624:	296351_1	296351_3	OK	24.0995	25.8372	0.100445	-0.55981	0.575608	0.986289	no
Fanca	Fanca	Fanca	chr8:1257:	296351_1	296351_3	OK	17.2512	18.006	0.061781	-0.35532	0.722347	0.998317	no
Fancb	Fancb	Fancb	chrX:1614:	296351_1	296351_3	OK	5.93487	6.16177	0.054128	-0.31254	0.754634	0.999381	no
Fancc	Fancc	Fancc	chr13:634:	296351_1	296351_3	OK	4.67393	4.79343	0.036425	-0.21864	0.826935	0.999381	no
Fancd2	Fancd2	Fancd2	chr6:1134:	296351_1	296351_3	OK	12.6089	12.6643	0.006326	-0.03744	0.970132	0.999381	no
Fance	Fance	Fance	chr17:284:	296351_1	296351_3	OK	7.93625	9.11051	0.199075	-1.64082	0.100834	0.645017	no
Fancf	Fancf	Fancf	chr7:5911:	296351_1	296351_3	OK	6.27976	5.74584	-0.12819	0.654314	0.512909	0.974194	no
Fancg	Fancg	Fancg	chr4:4301:	296351_1	296351_3	OK	12.4992	13.0912	0.066755	-0.47939	0.63166	0.987539	no
Lig4	Lig4	Lig4	chr8:9970:	296351_1	296351_3	OK	1.69222	1.79883	0.088142	-0.41978	0.674646	0.994929	no
Mre11a	Mre11a	Mre11a	chr9:1458:	296351_1	296351_3	OK	18.0932	17.659	-0.03504	0.209247	0.834255	0.999381	no
Mus81	Mus81	Mus81	chr19:548:	296351_1	296351_3	OK	12.4085	12.3272	-0.00949	0.056103	0.955259	0.999381	no
Nbn	Nbn	Nbn	chr4:1588:	296351_1	296351_3	OK	12.4697	11.7695	-0.08337	0.499245	0.617606	0.987539	no
Palb2	Palb2	Palb2	chr7:1292:	296351_1	296351_3	OK	5.62385	5.8394	0.054261	-0.31864	0.749999	0.999381	no
Prkdc	Prkdc	Prkdc	chr16:156:	296351_1	296351_3	OK	4.86084	4.86525	0.001311	-0.00773	0.993833	0.999381	no
Rad50	Rad50	Rad50	chr11:534:	296351_1	296351_3	OK	17.6204	17.1936	-0.03538	0.20091	0.840769	0.999381	no
Rad51	Rad51	Rad51	chr2:1189:	296351_1	296351_3	OK	32.5809	30.6392	-0.08865	0.522459	0.601351	0.987539	no
Rad51ap1	Rad51ap1	Rad51ap1	chr6:1268:	296351_1	296351_3	OK	27.0831	26.1599	-0.05004	0.300425	0.763853	0.999381	no
Rad52	Rad52	Rad52	chr6:1198:	296351_1	296351_3	OK	5.59764	5.33292	-0.06989	0.483419	0.628798	0.987539	no
Rad54b	Rad54b	Rad54b	chr4:1148:	296351_1	296351_3	OK	6.57015	6.96935	0.085099	-0.49385	0.621411	0.987539	no
Rad54l	Rad54l	Rad54l	chr4:1157:	296351_1	296351_3	OK	22.8249	22.9051	0.005061	-0.05405	0.956898	0.999381	no
Rbbp8	Rbbp8	Rbbp8	chr18:118:	296351_1	296351_3	OK	17.9269	18.9804	0.082385	-0.48674	0.626444	0.987539	no
Rpa1	Rpa1	Rpa1	chr11:751:	296351_1	296351_3	OK	70.7384	76.7909	0.118441	-0.57501	0.565285	0.98529	no
Rpa2	Rpa2	Rpa2	chr4:1323:	296351_1	296351_3	OK	47.5749	45.0404	-0.07898	0.457648	0.647206	0.989457	no
Rpa3	Rpa3	Rpa3	chr6:8205:	296351_1	296351_3	OK	63.6404	61.7963	-0.04242	0.253873	0.799594	0.999381	no
Trp53bp1	Trp53bp1	Trp53bp1	chr2:1209:	296351_1	296351_3	OK	10.5619	10.8512	0.038994	-0.22299	0.823546	0.999381	no
Xrcc1	Xrcc1	Xrcc1	chr7:2533:	296351_1	296351_3	OK	20.5435	21.1146	0.039555	-0.23783	0.812017	0.999381	no
Xrcc2	Xrcc2	Xrcc2	chr5:2519:	296351_1	296351_3	OK	8.10086	7.57964	-0.09595	0.569828	0.568795	0.98529	no
Xrcc3	Xrcc3	Xrcc3	chr12:1112:	296351_1	296351_3	OK	10.7956	9.69263	-0.15548	0.920976	0.357063	0.92973	no
Xrcc4	Xrcc4	Xrcc4	chr13:899:	296351_1	296351_3	OK	6.5307	8.17671	0.324284	-1.68872	0.091273	0.619395	no
Xrcc6	Xrcc6	Xrcc6	chr15:818:	296351_1	296351_3	OK	30.7905	29.1368	-0.07964	0.471761	0.637098	0.98787	no

Ctrl shRNA compare to BRD9 shRNA

test_id	gene_id	gene	locus	sample_1	sample_2	status	value_1	value_2	log2(fold_test_stat)	p_value	q_value	significant
Atm	Atm	Atm	chr9:5324:296352_1	296351_4	OK	3.77806	2.34076	-0.69067	2.92296	0.003467	0.060259	no
Atr	Atr	Atr	chr9:9575:296352_1	296351_4	OK	2.81796	2.16881	-0.37774	1.47409	0.140459	0.48596	no
Atrip	Atrip	Atrip	chr9:1089:296352_1	296351_4	OK	5.42063	5.22414	-0.05327	0.193525	0.846548	0.951207	no
Bard1	Bard1	Bard1	chr1:7107:296352_1	296351_4	OK	3.7382	3.61162	-0.0497	0.193856	0.846289	0.951207	no
Brca1	Brca1	Brca1	chr11:101:296352_1	296351_4	OK	8.96254	6.98399	-0.35986	1.59433	0.110862	0.433784	no
Brca2	Brca2	Brca2	chr5:1513:296352_1	296351_4	OK	2.17445	1.86012	-0.22525	1.14185	0.253518	0.624431	no
Brd9	Brd9	Brd9	chr13:740:296352_1	296351_4	OK	14.2411	6.86513	-1.0527	4.1548	3.26E-05	0.001523	yes
Chek1	Chek1	Chek1	chr9:3651:296352_1	296351_4	OK	8.1324	7.3928	-0.13756	0.557153	0.577423	0.849716	no
Chek2	Chek2	Chek2	chr5:1112:296352_1	296351_4	OK	12.035	9.86919	-0.28623	1.14088	0.253918	0.624542	no
Dclre1c	Dclre1c	Dclre1c	chr2:3341:296352_1	296351_4	OK	5.79928	6.00216	0.049607	-0.20203	0.839897	0.948552	no
Dna2	Dna2	Dna2	chr10:624:296352_1	296351_4	OK	7.49106	5.96492	-0.32867	1.34102	0.179915	0.540634	no
Fanca	Fanca	Fanca	chr8:1257:296352_1	296351_4	OK	20.6978	16.3323	-0.34175	1.57469	0.115327	0.441381	no
Fancb	Fancb	Fancb	chrX:1614:296352_1	296351_4	OK	5.18347	4.51905	-0.1979	0.731207	0.464653	0.785877	no
Fancc	Fancc	Fancc	chr13:634:296352_1	296351_4	OK	4.3104	4.88445	0.180373	-0.69436	0.487456	0.800117	no
Fancd2	Fancd2	Fancd2	chr6:1134:296352_1	296351_4	OK	5.04074	3.93173	-0.35847	1.40909	0.158808	0.510758	no
Fance	Fance	Fance	chr17:284:296352_1	296351_4	OK	14.7795	17.0259	0.204128	-1.2222	0.22163	0.593019	no
Fancf	Fancf	Fancf	chr7:5911:296352_1	296351_4	OK	3.96431	3.93918	-0.00917	0.028991	0.976872	0.939657	no
Fancg	Fancg	Fancg	chr4:4301:296352_1	296351_4	OK	6.84117	6.38881	-0.0987	0.376013	0.706907	0.908048	no
Lig4	Lig4	Lig4	chr8:9970:296352_1	296351_4	OK	3.44005	2.65875	-0.37168	1.33209	0.182831	0.543859	no
Mre11a	Mre11a	Mre11a	chr9:1458:296352_1	296351_4	OK	10.8731	11.3807	0.065831	-0.27523	0.78314	0.928451	no
Mus81	Mus81	Mus81	chr19:548:296352_1	296351_4	OK	13.2345	16.3519	0.305155	-1.24659	0.212548	0.582367	no
Nbn	Nbn	Nbn	chr4:1588:296352_1	296351_4	OK	19.1858	16.3024	-0.23496	1.01463	0.310283	0.674866	no
Palb2	Palb2	Palb2	chr7:1292:296352_1	296351_4	OK	2.64607	2.15615	-0.29539	0.998351	0.318109	0.680944	no
Prkdc	Prkdc	Prkdc	chr16:156:296352_1	296351_4	OK	1.88009	1.58982	-0.24194	0.959762	0.337175	0.698294	no
Rad50	Rad50	Rad50	chr11:534:296352_1	296351_4	OK	13.2912	11.6588	-0.18906	0.854259	0.392962	0.740451	no
Rad51	Rad51	Rad51	chr2:1189:296352_1	296351_4	OK	26.1799	24.9482	-0.06952	0.307236	0.758664	0.923513	no
Rad51ap1	Rad51ap1	Rad51ap1	chr6:1268:296352_1	296351_4	OK	18.7861	18.1586	-0.04901	0.203985	0.838365	0.948544	no
Rad52	Rad52	Rad52	chr6:1198:296352_1	296351_4	OK	6.38768	7.67091	0.264105	-1.57243	0.11585	0.442597	no
Rad54b	Rad54b	Rad54b	chr4:1148:296352_1	296351_4	OK	10.0349	8.57251	-0.22724	0.92007	0.357536	0.714234	no
Rad54l	Rad54l	Rad54l	chr4:1157:296352_1	296351_4	OK	17.1302	15.7953	-0.11704	0.933903	0.350354	0.70856	no
Rbbp8	Rbbp8	Rbbp8	chr18:118:296352_1	296351_4	OK	12.6825	11.2361	-0.1747	0.74938	0.453628	0.779699	no
Rpa1	Rpa1	Rpa1	chr11:751:296352_1	296351_4	OK	86.8846	74.6192	-0.21955	1.07461	0.282551	0.649676	no
Rpa2	Rpa2	Rpa2	chr4:1323:296352_1	296351_4	OK	42.1302	42.4016	0.009265	-0.04219	0.966347	0.990176	no
Rpa3	Rpa3	Rpa3	chr6:8205:296352_1	296351_4	OK	73.7534	99.6034	0.433485	-1.85318	0.063857	0.334515	no
Trp53	Trp53	Trp53	chr11:693:296352_1	296351_4	OK	128.726	134.766	0.06616	-0.38752	0.69837	0.904871	no
Xrcc1	Xrcc1	Xrcc1	chr7:2533:296352_1	296351_4	OK	32.9554	38.2928	0.21656	-0.98944	0.322449	0.683522	no
Xrcc2	Xrcc2	Xrcc2	chr5:2519:296352_1	296351_4	OK	5.27425	4.87411	-0.11383	0.427401	0.669087	0.893421	no
Xrcc3	Xrcc3	Xrcc3	chr12:112:296352_1	296351_4	OK	6.99467	6.2856	-0.15421	0.575316	0.565078	0.842691	no
Xrcc4	Xrcc4	Xrcc4	chr13:899:296352_1	296351_4	OK	6.45102	6.88508	0.093945	-0.32099	0.748219	0.92188	no
Xrcc5	Xrcc5	Xrcc5	chr1:7235:296352_1	296351_4	OK	18.0476	20.3343	0.172108	-0.75283	0.451553	0.779433	no
Xrcc6	Xrcc6	Xrcc6	chr15:818:296352_1	296351_4	OK	50.4383	47.4393	-0.08844	0.413835	0.678995	0.896406	no

4. The immunofluorescence (IF) images and experimentation presented in Figures 1/S1, 2/S2 have no positive controls (ie. knockdown of known regulators of HR) and in several instances only use a single BRD9 shRNAs. These experiments need to be strengthened significantly to support the authors suggestions.

Response: Thank you for this comment. We are now displaying positive controls (BRCA1 and 53BP1 knockdown) which we believe has strengthened the DR-GFP and IF data in **Figure 1B-J,2A** and **Figure S1A-C**. We also employed two shRNAs for the data presented in **Figure1** **Figure2** and **Figure S2**.

5. The resolution of the images presented in Fig. S2A is very poor making it nearly impossible to see the underlying numbers. Moreover, the cell cycle profiles themselves do not match the authors assertion that there is no significant effect on cell cycle profile following BRD9 knockdown. In fact, it is quite clear to see that there are proportionally fewer cells in G2/M (and possibly S) following BRD9 depletion. This is clearly an important point given that these DNA double-strand break repair mechanisms preferentially occur at certain points throughout the cell cycle. Shifting cell cycle dynamics as appears to be the case here could alter these mechanisms and their regulation indirectly.

Response: Thank you for this comment. We have repeated the cell cycle analysis 3 independent times. The cell cycle profile is now displayed with improved resolution. Moreover, we have quantified the percentage of cells in G1, S, and G2/M phases and provided this data in a color coded bar graph. As displayed in **supplementary figure 2A**, we do not see evidence of a significant change in the cell cycle profile following BRD9 knockdown.

6. *In Fig. S2B the authors present Western blot data that they claim is representative of the subcellular “chromatin” fraction. However, it cannot be concluded based on the presented data that this is in fact the case. The authors have not run cytoplasmic and/or nucleoplasmic fractions on these blots to demonstrate that they have in fact adequately fractionated protein samples. This should be done and control Western blots for proteins present in non-overlapping fractions should be included to support the validity of the claims.*

Response: Thanks for this suggestion. As suggested we have blotted both the cytoplasmic and nuclear fractions with GAPDH and Histone H3 antibodies, respectively. As displayed in **Fig.S2A** there is no significant overlap between cytoplasmic and chromatin binding fractions.

7. *In Fig. 4 the authors over-express GCN5 and PCAF in cells and demonstrate that acetylation levels of RAD54 increases in this setting. However, no conceptual rationale was presented as to why these particular enzymes were chosen. Are these enzymes actually the primary (and biologically relevant) mediators of RAD54 acetylation? Its hard to conclude based on the presented data (and lack of rationale) whether this is the case.*

Response: Thank you very much for your suggestion. We also screened for other key acetyltransferases like MOF, TIP-60 and P300 to assess if they can acetylate the RAD54 (new **Figure.S3E**). Data showed that GCN5 and PCAF, but not MO, TIP60 or P300 can mediate RAD54 acetylation.

Minor Points:

1. *The grammar throughout the manuscript is poor and should be improved to make the text easier to read/follow.*

Response: We have edited the manuscript for grammar.

2. *The authors focus their computational analysis of cancer mutational signatures in Figure 1 exclusively on ovarian cancer. Many other cancers, in particular BRCA1/2 mutated breast cancers, have overlapping mutational signatures (ie. Signature 3). It would be interesting to note whether or not the observations made here in ovarian cancer carry over to other Signature 3 tumours; or whether they are in fact specific to ovarian cancer (also see Major Point 1).*

Response: Thank you for this suggestion. We analyzed 42 BRD proteins in Breast cancer. The data shows that the mutations in the 6 bromodomain containing genes (BRD9, ASH1L, BRWD3, KMT2A, ZMYND8 and SMARCA4) associated with signature 3 in ovarian cancer were not associated with signature 3 in breast cancer, another tumor type enriched for signature 3.

Threshold for high HR mutation signature: 75%

	normal	mutation	p value
BRD4	0.249230769	0.3	0.474
TAF1L	0.252609603	0.148148148	0.9356
ATAD2	0.247156153	0.388888889	0.1363
BRD7	0.248212462	0.5	0.1684
KAT2A	0.249745158	0.25	0.6838
TRIM24	0.249743063	0.25	0.6097
ATAD2B	0.249740933	0.263157895	0.5346
BRD8	0.249230769	0.3	0.474
KAT2B	0.25102459	0.111111111	0.9256
TRIM66	0.250255363	0.166666667	0.8226
BAZ1A	0.246659815	0.5	0.05306
BRD9	0.247697032	0.5	0.1126
KMT2A	0.248427673	0.290322581	0.3637
ZMYND11	0.250254323	0	1
BAZ1B	0.250257467	0.214285714	0.72
BRDT	0.246421268	0.714285714	0.01253
PHIP	0.247156153	0.388888889	0.1363
ZMYND8	0.249486653	0.272727273	0.5448
BAZ2A	0.248197734	0.357142857	0.2566
BRPF3	0.25	0.222222222	0.7003
SMARCA2	0.252317199	0.071428571	0.9826
BPTF	0.247658689	0.333333333	0.2306
BRWD1	0.246861925	0.344827586	0.1621
SP100	0.250513347	0.181818182	0.804
BRD1	0.250511247	0.142857143	0.8672
BRWD3	0.249480249	0.260869565	0.5315
SP110	0.247959184	0.6	0.1027
BRD2	0.249745158	0.25	0.6838
CECR2	0.245087901	0.5	0.01827
SP140	0.249743063	0.25	0.6097
ASH1L	0.248421053	0.285714286	0.3709
EP300	0.247401247	0.347826087	0.193
BRD3	0.250511247	0.142857143	0.8672
CREEBBP		no mutation	
SP140L	0.251540041	0.090909091	0.9584
TRIM33	0.250513347	0.181818182	0.804
BAZ2B	0.25	0.24	0.6229

TRIM27	0.250254323	0	1
TAF1L	0.252609603	0.148148148	0.9356
PBRM1	0.251028807	0.153846154	0.8746
SMARCA4	0.251292658	0.166666667	0.8665

Reviewers' comments:

Reviewer #1 (Remarks to the Author):

The authors have addressed my questions, thus I recommend this manuscript for publication.

Reviewer #2 (Remarks to the Author):

The authors have responded to my comments adequately. The manuscript is acceptable for publication.

Reviewer #3 (Remarks to the Author):

The authors have largely dealt with my initial comments and concerns and have certainly improved the manuscript as a whole. However, I still have concerns related to their assertion that mutation of BRD9 is causing the associated Signature 3 mutational signature. I find it hard to conclude based on the data presented that mutation of BRD9 is the driving force behind the establishment of this mutational signature more broadly. I still believe the authors need to invest more experimental time into demonstrating whether or not mutation of BRD9 is a cause (not a consequence) of this mutational signature. However, I feel that if they can provide more compelling evidence to this end that this is an important study that will generate important additional questions and discussion within the field.

Reviewers' comments

Reviewer #1 (Remarks to the Author):

The authors have addressed my questions, thus I recommend this manuscript for publication.

RESPONSE: Thank you, again, for your constructive feedback and review

Reviewer #2 (Remarks to the Author):

The authors have responded to my comments adequately. The manuscript is acceptable for publication.

RESPONSE: Thank you, again, for your constructive feedback and review

Reviewer #3 (Remarks to the Author):

The authors have largely dealt with my initial comments and concerns and have certainly improved the manuscript as a whole. However, I still have concerns related to their assertion that mutation of BRD9 is causing the associated Signature 3 mutational signature. I find it hard to conclude based on the data presented that mutation of BRD9 is the driving force behind the establishment of this mutational signature more broadly. I still believe the authors need to invest more experimental time into demonstrating whether or not mutation of BRD9 is a cause (not a consequence) of this mutational signature. However, I feel that if they can provide more compelling evidence to this end that this is an important study that will generate important additional questions and discussion within the field.

RESPONSE: Thank you, again, for your detailed review and constructive criticism. Actually, what we mentioned in the manuscript is that BRD9 mutation is associated with signature 3. In order to address your concern, we have performed several additional analyses in TCGA in not only ovarian cancer, but also other cancers highly enriched for HR deficiency and Signature 3. Stratton and colleagues have previously reported that Signature 3 is strongly associated with BRCA1 and BRCA2 mutations within individual cancer types with nearly all cases of BRCA1 and BRCA2 associated cancers showing large contributions from signature 3. BRCA1 and BRCA2 mutations are by far the most common source of HR deficiency in ovarian cancer. If BRD9 were a consequence of the Signature 3 mutational signature (i.e. a passenger mutation), we would have expected to have identified a significant fraction of the BRD9 mutations identified to also be associated with mutations in either BRCA1 or BRCA2 mutations. However, we found that mutations in BRCA1, BRCA2, and BRD9 were mutually exclusive in ovarian cancer with a high contribution of Signature 3 (75% threshold). We also extended these studies to include breast cancer, pancreatic cancer, and prostate cancer as a subset of these tumors are also known to be HR deficient. In all cases, the BRD9 mutations were mutually exclusive of

mutations in BRCA1 or BRCA2. The relationship of expression of Signature 3 and BRD9 mutation in 4 cancer type patients with/without BRCA1/2 mutation was also investigated. The excluding of BRCA1/2 mutation patients does not affect the association of BRD9 mutation and high Signature3, which means BRD9 mutation is still associated with signature 3 in BRCA1/2 wild type samples. These data suggest that BRD9 mutations are associated with Signature 3 and additionally, might not be passenger mutations of Signatures. We have provided this data below and as supplementary Figure 10A-B. Again, these analysis simply give us a hint of a potential role of BRD9 in HR, to establish a casual role of BRD9 in HR, we performed all the functional studies in the manuscript. These data collectively support a role of BRD9 in HR.

Thank you, again, for your constructive feedback on our manuscript.

1. Association of BRCA1/2 mutations and BRD9 mutations in ovarian, breast, pancreatic, and prostate cancer with signature 3(75% threshold).

Ovarian Cancer

BRCA1/2 \ BRD9	No mutation	Mutation
No mutation	92	9
Mutation	8	0

Breast Cancer

BRCA1/2 \ BRD9	No mutation	Mutation
No mutation	227	15
Mutation	4	0

Pancreatic Cancer

BRCA1/2 \ BRD9	No mutation	Mutation
No mutation	21	0
Mutation	0	0

Prostate Cancer

BRCA1/2		
BRD9	No mutation	Mutation
No mutation	123	1
Mutation	0	0

2. Association of Signature 3 and BRD9 mutation in 4 cancer type patients with/without BRCA1/2 mutation

All Patients (Fisher's Exact Test for Count Data, p-value = 0.0001693)

Sig.3		
BRD9	Low	High
No mutation	1563	510
Mutation	7	13

Patients without BRCA1/2 mutation (Fisher's Exact Test for Count Data, p-value = 0.001187)

Sig.3		
BRD9	Low	High
No mutation	1522	499
Mutation	7	11

REVIEWERS' COMMENTS:

Reviewer #3 (Remarks to the Author):

I am happy that the authors have taken my previous comments on board and that they have toned down their association of BRD9 mutation with the Signature 3 mutational signature. It appears largely convincing that mutation of BRD9 is associated with HR deficient cancers outside of a BRCA1/S mutation context. This suggests a role for BRD9 function in HR which is bourn out in the manuscript.